# Coordinative metabolism of glutamine carbon and nitrogen in proliferating cancer cells under hypoxia

Yuanyuan Wang[1,4], Changsen Bai[1], Yuxia Ruan[1], Miao Liu[1], Qiaoyun Chu[2], Li Qiu[1], Chuanzhen Yang[2] & Binghui Li[1,2,3]

Under hypoxia, most of glucose is converted to secretory lactate, which leads to the overuse of glutamine-carbon. However, under such a condition how glutamine nitrogen is disposed to avoid over-accumulating ammonia remains to be determined. Here we identify a metabolic flux of glutamine to secretory dihydroorotate, which is indispensable to glutamine-carbon metabolism under hypoxia. We found that glutamine nitrogen is necessary to nucleotide biosynthesis, but enriched in dihyroorotate and orotate rather than processing to its downstream uridine monophosphate under hypoxia. Dihyroorotate, not orotate, is then secreted out of cells. Furthermore, we found that the specific metabolic pathway occurs in vivo and is required for tumor growth. The identified metabolic pathway renders glutamine mainly to acetyl coenzyme A for lipogenesis, with the rest carbon and nitrogen being safely removed. Therefore, our results reveal how glutamine carbon and nitrogen are coordinatively metabolized under hypoxia, and provide a comprehensive understanding on glutamine metabolism.

[1] Department of Cancer Cell Biology, Tianjin's Key Laboratory of Cancer Prevention and Therapy, National Clinical Research Center for Cancer, Tianjin Medical University Cancer Institute and Hospital, 300060 Tianjin, China. [2] Department of Biochemistry and Molecular Biology, Capital Medical University, 100069 Beijing, China. [3] Advanced Innovation Center for Human Brain Protection, Capital Medical University, 100069 Beijing, China. [4] Present address: Center of Diagnosis and Treatment of Breast Disease, the Affiliated Hospital of Qingdao University, 266071 Qingdao, China. Co-first authors: Yuanyuan Wang, Changsen Bai, Yuxia Ruan. Correspondence and requests for materials should be addressed to B.L. (email: bli@ccmu.edu.cn)

Proliferating cancer cells comprehensively rewire their metabolism to sustain growth and survival in the harsh conditions, such as hypoxia and nutrition deficiency[1]. Upon the resurgence of research interest into cancer metabolism, aberrant glucose utilization has been centrally studied recently. As a famous hallmark of cancers, aerobic glycolysis, termed the Warburg effect, is characterized by the increased metabolic flux of glucose to secretory lactate[2]. This process leads to the lack of carbon source from glucose to make building bricks, especially lipids, for cell proliferation. Therefore, the alternative carbon source is required for cell growth. Second to glucose, glutamine, the most abundant amino acid in the human blood[3], can serve as a ready source of carbon to support energy generation and biomass accumulation.

Glutamine plays a pleiotropic role in cellular functions[4]. Directly, glutamine can be incorporated to protein, and regulate protein translation and trafficking[5]. Through catabolic transformations, glutamine provides carbon and nitrogen for the biosynthesis of non-essential amino acids[5] and nucleotides[6,7]. In addition, glutamine can also forward fuel the citric acid cycle (CAC)[8,9]. Under hypoxia, the glutamine consumption in proliferating cells is elevated, and it preferentially provides carbon for fatty acid biosynthesis through reductive carboxylation[10], by which glutamine-derived α-ketoglutarate is reduced to citric acid by isocitrate dehydrogenases with NADPH oxidizing to $NADP^+$.

One glutamine contains five carbon atoms and two nitrogen atoms in the forms of amine and amide groups. When cells begin to addict to glutamine carbon, which usually happens on proliferating cancer cells under hypoxia[4], how do they deal with the potentially overflowed nitrogen? It has long been supposed that glutamine offers α-ketoglutarate for cells by deamination through glutaminase (GLS)[11] and glutamate dehydrogenase (GLUD)[9]. Concomitantly with these processes, the increasing amount of ammonia is produced and could be toxic to cells[12,13]. Although a recent report showed that breast cancer cells could slightly recycle ammonia to generate amino acids through GLUD[14], GLUD-mediated conversion of ammonia and α-ketoglutarate to glutamate does not efficiently occur in most of cancer cells[4,15]. To avoid over-accumulating ammonia, the best way for proliferating cancer cells is to reduce its generation. Therefore, how glutamine nitrogen is coordinatively metabolized to avoid releasing ammonia deserves to be further determined.

Different elements in a metabolite usually have different metabolic fates, thus their coordinative metabolism is critical to maintain the metabolic homeostasis in cells. Once the changed microenvironment perturbs the homeostasis, re-building a new coordinative metabolism is required. Here we show that hypoxia alters glutamine metabolism and drives a new metabolic homeostasis of its carbon and nitrogen.

## Results

**Requirement of glutamine-nitrogen for cell survival.** Glutamine is required for cell survival[16–19], and its loss induced cell death (Supplementary Fig. 1a). Supplementation with nucleosides, but not α-ketoglutarate and non-essential amino acids including glutamate, significantly suppressed cell death in MCF-7, HeLa, and A549 cells induced by glutamine loss (Supplementary Fig. 1a–1c), supporting the well-established notion that glutamine is necessary for nucleotide biosynthesis[6]. In fact, glutamine can be potentially synthesized from glutamate by glutamine synthetase (GS) (Supplementary Fig. 2a). However, glutamine deprivation led to a dramatic loss of cellular glutamine (about 5% of the control) but showed no or less effect on other non-essential amino acids and the intermediates in the CAC in MCF-7 and HeLa cells (Supplementary Fig. 2b, c). Notably, the culture

medium did not contain non-essential amino acids including glutamate. It suggests that cells could synthesize glutamate from α-ketoglutarate (Supplementary Fig. 2a). We then used the labeled carbon source, $^{13}C_6$-glucose, to culture MCF-7 and HeLa cells, and the $^{13}C$ tracing analysis showed that α-ketoglutarate and glutamate were substantially labeled by $^{13}C$ even in the presence of glutamine but the glucose-derived fraction significantly increased in the absence of glutamine (Supplementary Fig. 2d). Nonetheless, glutamine was not labeled at all in the presence of glutamine but slightly labeled, when compared to α-ketoglutarate and glutamate, in the absence of glutamine (Supplementary Fig. 2d). These results suggest that glutamine cannot be efficiently synthesized in cells even upon its scarcity, and it could be attributed to the low level of GS. We then over-expressed GS in MCF-7 cells (Supplementary Fig. 1d), and found it dramatically increased the labeling of glutamine, but not glutamate and α-ketoglutarate, by $^{13}C_6$-glucose (Supplementary Fig. 2e). GS expression also inhibited cell death and restored cell proliferation upon supplementation with glutamate, α-ketoglutarate or pyruvate in MCF-7, HeLa, and A549 cells in the absence of glutamine (Supplementary Fig. 1e). Conversely, GS knockdown sensitized cells to glutamine loss independently of supplementation of nutrients (Supplementary Fig. 1f). These data suggest that glutamine amide-nitrogen is indispensable to cell proliferation. When glutamine nitrogen is used, its carbon, if beyond biomass accumulation, could be oxidized via the CAC. However, when the cellular requirement of glutamine carbon is increased in some conditions, such as under hypoxia, how do cells metabolically dispose of the overflowed nitrogen?

**Hypoxia increases glutamine uptake and decreases ammonia production.** Under hypoxia, glutamine was used as the major carbon source, especially for lipid biosynthesis (Fig. 1a)[10]. Indeed, we detected a significantly increased uptake of glutamine but decreased excretion of glutamate in MCF-7, HeLa, and 4T1 cells (Fig. 1b and Supplementary Fig. 3a), suggesting an elevated intracellular utilization of glutamine-carbon under hypoxia. We then traced the metabolic flux of U-$^{13}C$-labeled glutamine. Consistent with the previous report[10], the fraction of glutamine-derived acetyl-CoA $m + 2$, the precursor of lipogenesis, was significantly enhanced (Fig. 1c). In addition, the increased enrichment of glutamine-$^{13}C$ in α-ketoglutarate, citrate, glutamate, and proline, downstream metabolites of glutamine, was also detected (Fig. 1d, e and Supplementary Figs. 4 and 5). These data suggest the increased uptake of glutamine as the carbon source under hypoxia. Next, we tried to figure out the metabolic fate of glutamine nitrogen under hypoxia. Glutamine can be deaminated and release its nitrogen as ammonia[4] (Fig. 1a), thus we measured the ammonia in the culture medium. As showed in Fig. 1f, the production of ammonia was actually significantly decreased. Ammonia can be removed by forming urea in human body[20,21], and we did not detect urea in the culture medium but measured a decreased level of intracellular urea (Fig. 1g). In addition, we also detected a decreased excretion of alanine, the potential ammonia carrier[21], in the medium (Supplementary Fig. 3b). Therefore, glutamine-nitrogen should be enriched in some other cellular metabolites under hypoxia.

**Accumulation of cellular nucleotide precursors under hypoxia.** We then used HeLa and 4T1 cell lines as the models to screen the nitrogen metabolome including 226 nitrogen-contained metabolites. Hypoxia significantly ($P < 0.05$) increased six and decreased 11 overlapped metabolites in both HeLa and 4T1 cells (Fig. 2a). Interestingly, five increased compounds, including dihydroorotate, orotate, IMP, guanosine, and inosine, were involved in the

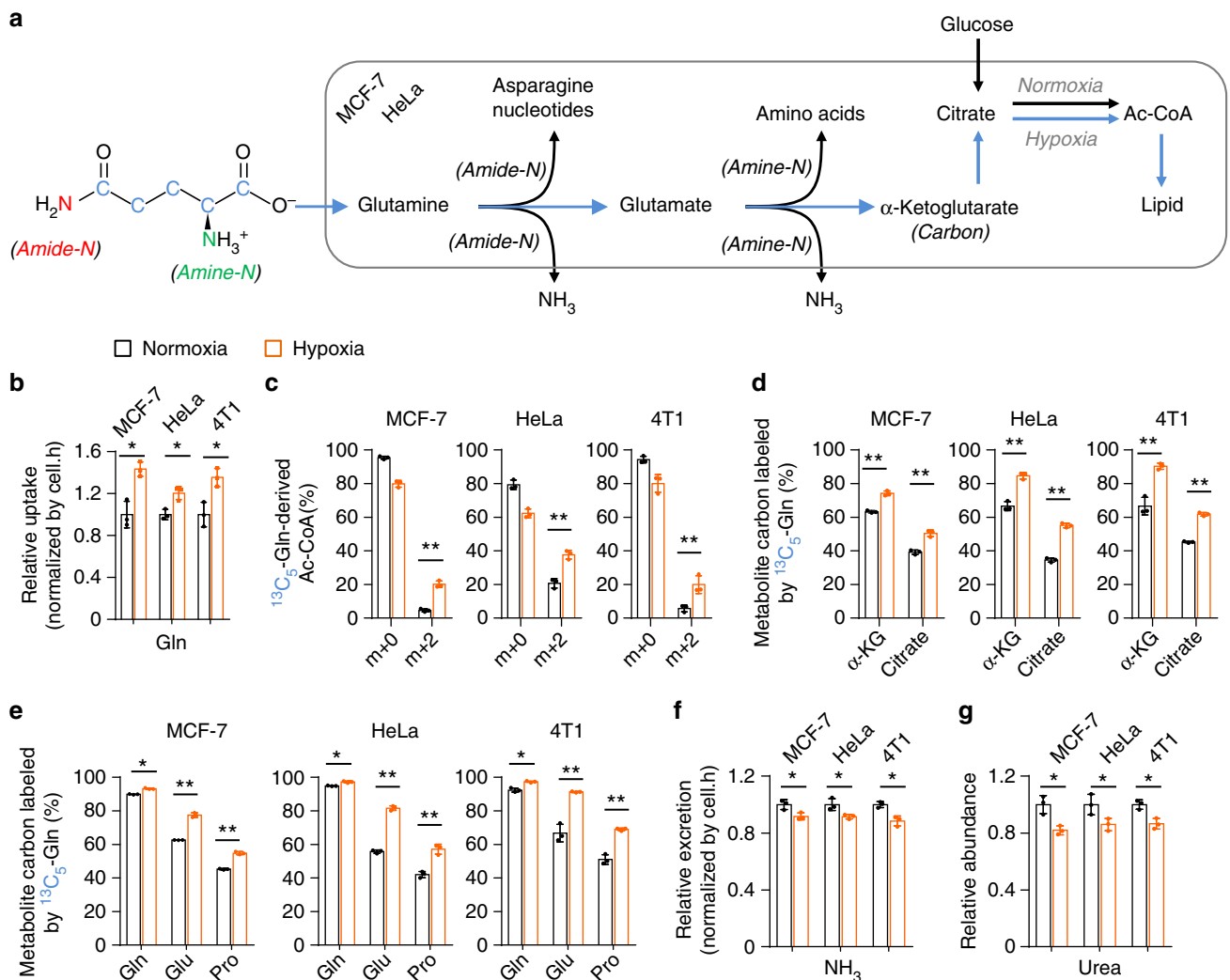

**Fig. 1** Increased glutamine as the carbon source under hypoxia. **a** A schematic to show the metabolism of glutamine carbon and nitrogen. **b** Relative glutamine uptake in MCF-7, HeLa, and 4T1 cells cultured under hypoxia and normoxia for 8 h. **c** Mass isotopomer analysis of acetyl-CoA in MCF-7, HeLa, and 4T1 cells cultured with the medium containing 1 mM of $^{13}C_5$-glutamine under hypoxia and normoxia for 8 h. **d, e** The $^{13}C_5$-labeled fraction of metabolites in MCF-7, HeLa, and 4T1 cells cultured with the medium containing 1 mM of $^{13}C_5$-glutamine for 8 h under hypoxia or normoxia. **f** Relative ammonia excretion from MCF-7, HeLa, and 4T1 cells cultured under hypoxia and normoxia for 8 h. **g** Relative cellular urea in MCF-7, HeLa, and 4T1 cells cultured under hypoxia and normoxia for 8 h. All cultures were supplied with 10% dialyzed serum. Values are the means ± SEM of three independent experiments. *$p < 0.05$; **$p < 0.01$ (Student's $t$-test)

nucleotide biosynthesis pathway (Fig. 2b). We verified again that IMP, the precursor of AMP and GMP, significantly enhanced and carbamoyl-Asp, dihydroorotate, and orotate, the precursors of UMP, dramatically increased in HeLa, MCF-7, and 4T1 cells under hypoxia (Fig. 2c, d). Furthermore, we measured the cellular nucleotides and their derivatives in HeLa and MCF-7 cells under hypoxia. As shown in Fig. 2e, only the cellular GMP raised upon hypoxia. On the contrary, UMP and its derivatives, UDP, UTP, and CTP, even reduced under hypoxia (Fig. 2e), which was also supported by the metabolomic profiling that UTP and thymidine triphosphate (dTTP), the other UMP derivative, were observed to decrease in the hypoxic cells (Fig. 2a).

Surprisingly, the metabolomic analysis showed that aspartate, the major precursor for carbamoyl-Asp, dihydroorotate, and orotate (Fig. 2b), significantly declined under hypoxia (Fig. 2a), which was confirmed again in MCF-7, HeLa, and 4T1 cells (Fig. 2d). We further measured the levels of cellular amino acids in MCF-7 and HeLa cells under hypoxia and normoxia. We observed the increased cellular glutamine and glutamate in both

cell lines under hypoxia (Supplementary Fig. 6), which should result from the increased uptake of glutamine (Fig. 1b). Intriguingly, our results in MCF-7 and HeLa cells clearly indicated that only aspartate significantly decreased in both cell lines (Supplementary Fig. 6). This could explain why accumulated IMP (Fig. 2c) led to an increase in GMP but not in AMP (Fig. 2e), because the conversion of IMP to AMP required aspartate participation (Fig. 2b).

Taken together, these data indicate that hypoxia substantially lead to the accumulation of cellular nucleotide precursors, in particular pyrimidine precursors including carbamoyl-Asp, dihydroorotate, and orotate.

**Glutamine-nitrogen is enriched in dihydroorotate and orotate under hypoxia.** Now we traced the assimilation of glutamine-nitrogen using amide-$^{15}N$-labeled or amine-$^{15}N$-labeled glutamine in HeLa and MCF-7 cells. Glutamine amine-$^{15}N$ can be assimilated into both purine and pyrimidine nucleotides with one

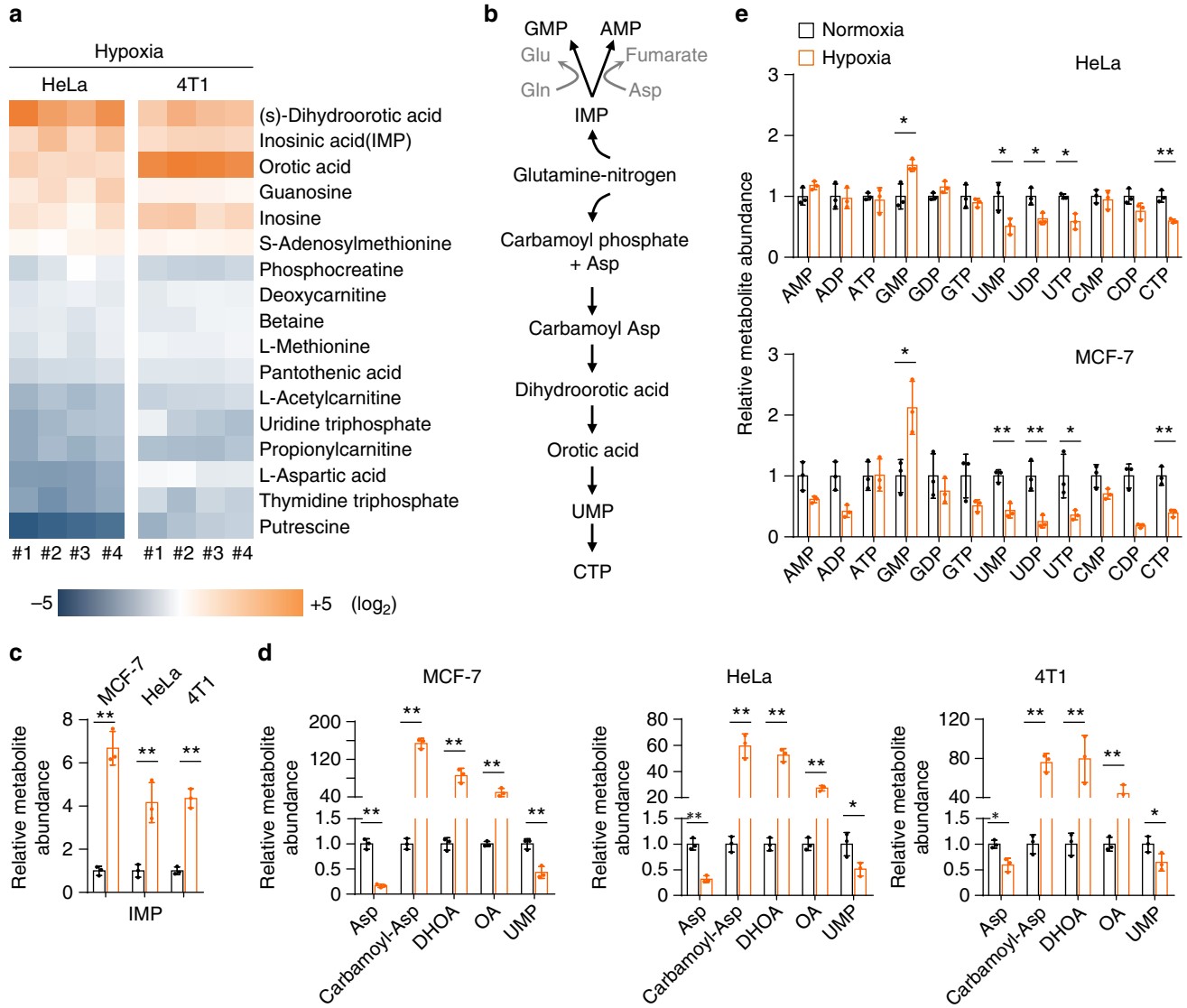

**Fig. 2** Accumulation of cellular nucleotide precursors under hypoxia. **a** Heatmap of N-contained metabolites in HeLa and 4T1 cells significantly ($p < 0.05$) affected by hypoxia for 8 h. Cellular metabolites were measured by LC–MS-based metabolomics. **b** A schematic to show the metabolic assimilation of glutamine-nitrogen to nucleotide biosynthesis. **c** Relative cellular IMP in MCF-7, HeLa, and 4T1 cells cultured under hypoxia and normoxia for 8 h.
**d** Relative cellular aspartate, carbamoyl-aspartate, dihydroorotate, orotate, and UMP in MCF-7, HeLa, and 4T1 cells cultured under hypoxia and normoxia for 8 h. **e** Relative cellular nucleotides in MCF-7 and HeLa cells cultured under hypoxia and normoxia for 8 h. All cultures were supplied with 10% dialyzed serum. Values are the means ± SEM of three independent experiments. *$p < 0.05$; **$p < 0.01$ (Student's $t$-test)

$^{15}$N atom ($m + 1$) in aspartate, dihydroorotate, orotate, UMP, and IMP (the precursor of purine nucleotides) (Fig. 3a and Supplementary Fig. 7). In contrast, glutamine amide-$^{15}$N labeled dihydroorotate, orotate, and UMP with one $^{15}$N atom ($m + 1$) and IMP with two $^{15}$N atoms ($m + 2$) (Fig. 3a and Supplementary Fig. 8). The labeled fractions of dihydroorotate and orotate significantly increased under hypoxia (Fig. 3b, c), suggesting an enrichment of glutamine-nitrogen in dihydroorotate and orotate under hypoxia. However, dihydroorotate and orotate inefficiently processed to its downstream UMP, because cellular UMP was less labeled by glutamine-$^{15}$N (Fig. 3b, c). The reduced labeled fraction of IMP was also reduced under hypoxia (Fig. 3b, c), suggesting a decrease in IMP biosynthesis, which possibly resulted from the slowed cell proliferation rate in view of the accumulation of cellular IMP (Fig. 2c).

Glutamine-amine-$^{15}$N labeled non-essential amino acids, such as glutamate, proline, asparagine, aspartate, and alanine (Fig. 3d and Supplementary Fig. 9). In contrast, glutamine-amide-$^{15}$N

predominantly labeled asparagine in both MCF-7 and HeLa cells (Fig. 3e and Supplementary Fig. 10). In addition, glutamine-amide-$^{15}$N also slightly labeled non-essential amino acids in MCF-7 breast cancer cells (Fig. 3e), consistent with a recent report showing that breast cancer cells could metabolically recycle ammonia released from glutamine-amide nitrogen[14]. However, almost all the $^{15}$N-labeled fraction of amino acids decreased under hypoxia (Fig. 3d, e). Taken together, these data suggest that glutamine nitrogen is enriched in dihydroorotate and orotate but not in amino acids under hypoxia.

**Promotion of aspartate to pyrimidine precursors under hypoxia.** Glutamine carbon can be potentially integrated into pyrimidine nucleotides after it has been converted to aspartate through the CAC-mediated oxidative pathway or α-ketoglutarate carboxylation reductive pathway (Fig. 3a). The two pathways can be distinguished by determining the enrichment of $^{13}C_5$-glutamine-derived $^{13}$C in acetyl-CoA, aspartate, citrate,

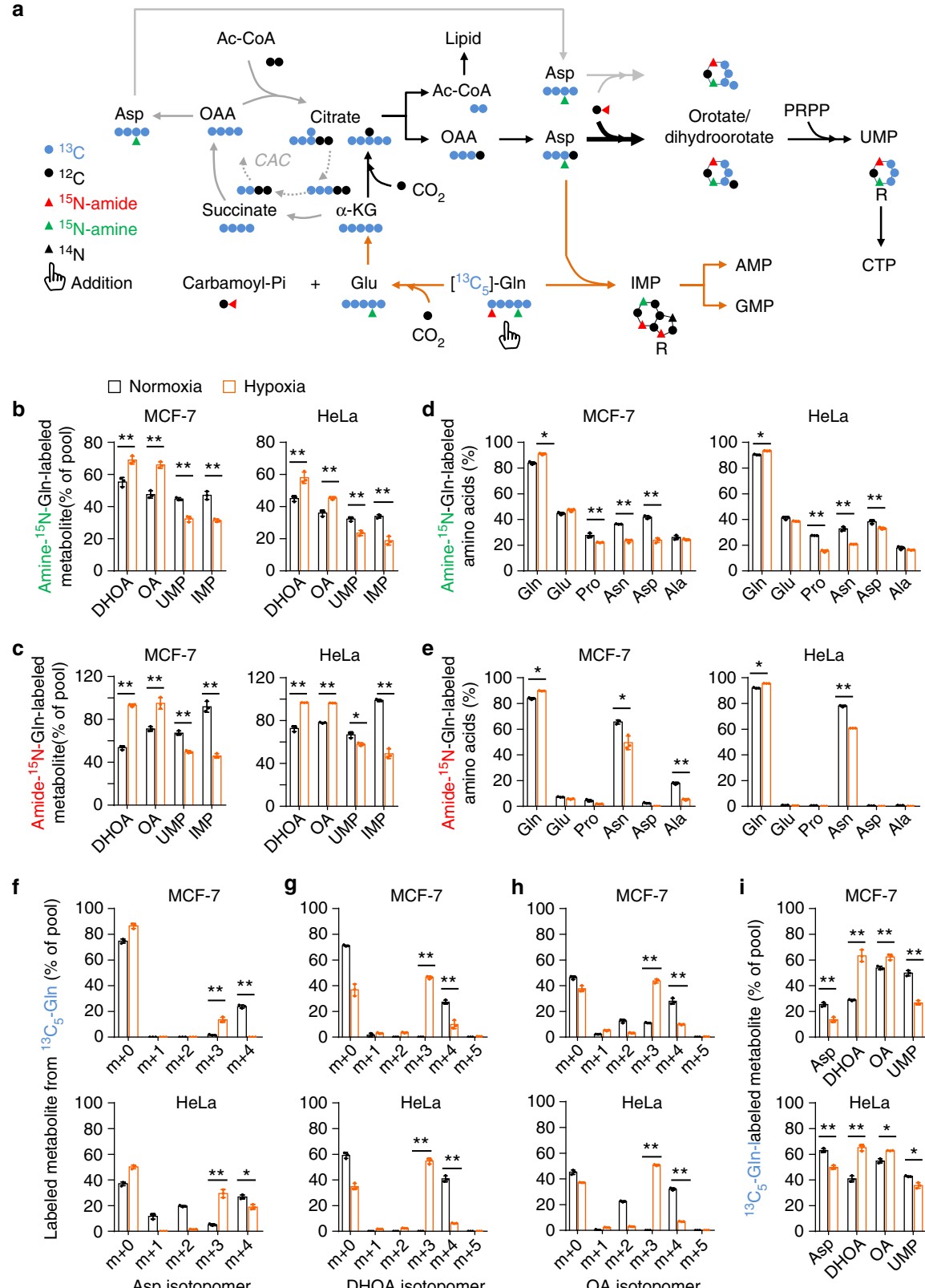

dihydroorotate, and orotate. Normally, glutamine-$^{13}C$ entered the CAC, and then was enriched in aspartate $m + 4$ (Fig. 3f), citrate $m + 4$ (Supplementary Fig. 4), dihydroorotate $m + 4$ and orotate $m + 4$ (Fig. 3g, h). Under hypoxia, $^{13}C_5$-glutamine-derived acetyl-CoA $m + 2$ (Fig. 1c), aspartate $m + 3$ (Fig. 3f) and citrate $m + 5$ (Supplementary Fig. 4), dihydroorotate $m + 3$, and orotate $m + 3$

(Fig. 3g, h) were generated via the reductive pathway. Finally, UMP $m + 3$ was produced in both pathways (Fig. 3a and Supplementary Fig. 11). These results clearly showed that glutamine carbon was integrated into acetyl-CoA (the precursor for lipid biosynthesis) and aspartate-derived dihydroorotate and orotate through the reductive pathway under hypoxia. Moreover, the

**Fig. 3** Metabolic flux of glutamine-nitrogen and glutamine-carbon in nucleoside biosynthesis. **a** A schematic to show the metabolism of isotope-labeled glutamine. **b**, **c** The $^{15}$N-labeled fraction of dihydroorotate, orotate, UMP, and IMP in MCF-7 and HeLa cells cultured with the medium containing 1 mM of amine-$^{15}$N-glutamine or amide-$^{15}$N-glutamine for 8 h under hypoxia or normoxia. **d**, **e** The $^{15}$N-labeled fraction of amino acids in MCF-7 and HeLa cells cultured with the medium containing 1 mM of amine-$^{15}$N-glutamine or amide-$^{15}$N-glutamine for 8 h under hypoxia or normoxia. f–h Mass isotopomer analysis of aspartate, dihydroorotate, and orotate in MCF-7 and HeLa cells cultured with the medium containing 1 mM of $^{13}$C$_5$-glutamine for 8 h under hypoxia or normoxia. **i** The $^{13}$C$_5$-labeled fraction of metabolites in MCF-7 and HeLa cells cultured with the medium containing 1 mM of $^{13}$C$_5$-glutamine for 8 h under hypoxia or normoxia. All cultures were supplied with 10% dialyzed serum. Values are the means ± SEM of three independent experiments. *$p <$ 0.05; **$p <$ 0.01 (Student's $t$-test)

fraction of glutamine-$^{13}$C-labeled acetyl-CoA, dihydroorotate, and orotate significantly increased (Figs. 1c and 3i). These data suggest that the biosyntheses of acetyl-CoA, dihydroorotate, and orotate from glutamine were urged by hypoxia.

However, aspartate, the direct precursor of dihydroorotate, was less efficiently labeled by glutamine-$^{13}$C under hypoxia (Fig. 3i). This most likely resulted from the multienzyme complexes involved in the pyrimidine biosynthesis where the newly synthesized aspartate by cytosolic glutamic-oxaloacetate transaminase 1 (GOT1) can be efficiently converted to dihydroorotate and orotate[4,22]. To test this speculation, we cultured HeLa and MCF-7 cells with $^{13}$C$_4$,$^{15}$N-labeled aspartate, and traced the isotope-labeled intermediates in the pyrimidine biosynthesis. Both cell lines substantially absorbed exogenous $^{13}$C$_4$,$^{15}$N-labeled aspartate $m + 5$ (Fig. 4a, b), and we also detected the newly synthesized $^{13}$C$_4$-labeled aspartate $m + 4$, $^{15}$N-labeled aspartate $m + 1$ and $^{15}$N-labeled glutamate $m + 1$ (Fig. 4a, b). $^{13}$C$_4$-aspartate $m + 4$ carried glutamine-derived amine nitrogen, while $^{15}$N-aspartate $m + 1$ contained $^{13}$C$_4$,$^{15}$N-aspartate-derived amine nitrogen mediated by $^{15}$N-glutamate $m + 1$ (Fig. 4c). Hypoxia increased the uptake and biosynthesis of aspartate (Fig. 4a, b). Interestingly, in the normal condition, aspartate inefficiently labeled dihydroorotate, orotate, and UMP (Fig. 4a, b). However, hypoxia promoted the labeling of dihydroorotate and orotate, but not UMP, by $^{13}$C$_4$,$^{15}$N-aspartate $m + 5$ and $^{13}$C$_4$-aspartate $m + 4$. In particular, the newly synthesized $^{13}$C$_4$-aspartate $m + 4$ was more efficiently incorporated into dihydroorotate and orotate than the absorbed $^{13}$C$_4$,$^{15}$N-aspartate $m + 5$. These data are consistent with the notion of metabolic multienzyme complexes and support that hypoxia strongly boosts entry of aspartate to dihydroorotate and orotate, which possibly leads to the decreased cellular aspartate (Fig. 2d).

**Association of glutamine-carbon metabolism with its nitrogen assimilation under hypoxia**. To test whether the increased metabolic flux of glutamine to dihydroorotate and orotate is required for cell survival under hypoxia, we knockdowned the involved enzymes, such as GOT1, carbamoyl-phosphate synthetase 2, aspartate transcarbamylase (ATCase) and dihydroorotase (CAD) and dihydroorotate dehydrogenase (DHODH) (Fig. 5a), in MCF-7 and HeLa cells (Fig. 5b and Supplementary Fig. 12a). DHODH and CAD were the key enzymes involved in the biosynthesis of pyrimidine nucleotides, thus knockdown of DHODH and CAD apparently suppressed proliferation of these cells even in the normal condition (Fig. 5b and Supplementary Fig. 12a). In contrast, GOT1 knockdown slightly affected cell proliferation (Fig. 5b and Supplementary Fig. 12a). However, knockdown of CAD or GOT1 strongly, while DHODH knockdown marginally, sensitized cancer cells to hypoxia (Fig. 5b and Supplementary Fig. 12a). These results suggest that the biosynthesis of dihydroorotate, not orotate, is indispensable to survival under hypoxia.

To survive hypoxia, cells could excrete the accumulated metabolites. Thus, we measured the excretion of carbamoyl aspartate, dihydroorotate, and orotate in the culture medium. No

carbamoyl aspartate was detected under both hypoxia and normoxia. However, we observed a significantly increased amount of dihydroorotate, but not orotate, in hypoxic medium of various cell lines, such as MCF-7, HeLa, A549, HCC-LM3, SGC-7901, and 4T1 (Fig. 5c), suggesting it could be an universal phenomenon responding to hypoxia. The overall conversion of glutamine and dioxide carbon to acetyl-CoA and secretory dihydroorotate, not orotate, consumes electrons (Supplementary Fig. 13), which could remit hypoxia-induced electron accumulation. The reprogrammed metabolic pathway essentially renders glutamine only to acetyl-CoA for lipogenesis under hypoxia, and glutamine-amide and glutamine-amine groups are incorporated into secretory dihydroorotate by CAD and GOT1. These observations could explain why hypoxia increased the utilization of glutamine-carbon but decreased the release of ammonia (Fig. 1). This speculation was further supported by the results that knockdown of CAD or GOT1 enhanced the production of ammonia under hypoxia (Fig. 5d).

As expected, knockdown of CAD or GOT1 dramatically suppressed hypoxia-induced dihydroorotate and orotate (Fig. 5e, f). Meantime, their depletion was also found to reduce glutamine-derived α-ketoglutarate, citrate, and acetyl-CoA (Fig. 5g and Supplementary Fig. 12b, c). Moreover, supplementation with aspartate did not effectively restore the accumulation of dihydroorotate and orotate even in cells with GOT1 depletion (Fig. 5e, f), although the absorbed aspartate can be used to synthesize dihydroorotate and orotate under hypoxia (Fig. 4a, b). These data suggest that glutamine-carbon metabolism is associated with its nitrogen assimilation to dihydroorotate. To further confirm this speculation, we treated HeLa cells with α-ketoglutarate, the carbon form of glutamine. Our results showed that α-ketoglutarate supplementation reduced glutamine uptake (Supplementary Fig. 12d), and almost completely scavenged the accumulated cellular dihydroorotate and orotate (Fig. 5h, i) and suppressed the excretion of dihydroorotate under hypoxia (Fig. 5j). Supplementation with uridine and/or α-ketoglutarate increased proliferation rate of HeLa/shCAD and HeLa/shGOT1 cells in the normal condition (Supplementary Fig. 12e), but uridine alone did not while α-ketoglutarate alone or in combination with uridine substantially restored cell proliferation under hypoxia (Fig. 5k). Taken together, our results suggest that the metabolism of glutamine-carbon is necessary to cell survival and depends on the increased biosynthesis of dihydroorotate under hypoxia.

**Hypoxia-induced NADH accumulation promotes biosynthesis and excretion of dihydroorotate**. Next, we investigated how hypoxia promoted the biosynthesis and excretion of dihydroorotate. We measured the protein levels of related enzymes, including GOT1, CAD, DHODH, and uridine monophosphate synthetase (UMPS) (Fig. 5e), as well as HIF-1α. As a typical indicator, HIF-1α was indeed induced by hypoxia, but the expression of these metabolic enzymes were not enhanced (Fig. 6a). CAD is the critical enzyme for the biosynthesis of dihydroorotate, and can be activated by phosphorylation[23,24].

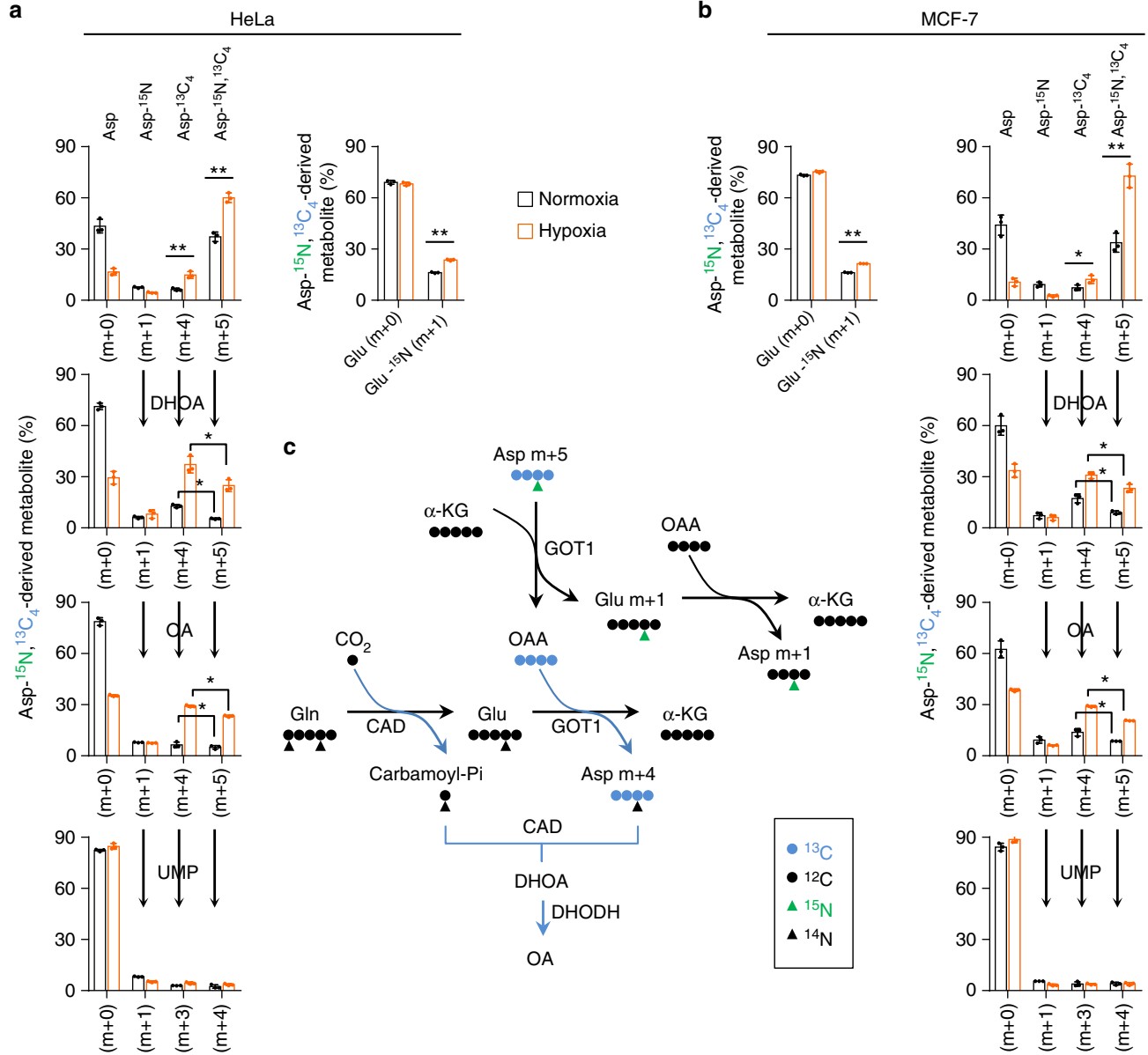

**Fig. 4** Promotion of aspartate to pyrimidine precursors under hypoxia. **a**, **b** Mass isotopomer analysis of aspartate, glutamate, dihydroorotate, orotate, and UMP in MCF-7 and HeLa cells cultured with the medium containing 10 mM of $^{13}C_4$, $^{15}N$-aspartate for 8 h under hypoxia or normoxia. Values are the means ± SEM of three independent experiments. *$p < 0.05$; **$p < 0.01$ (Student's $t$-test). **c** A schematic to show the metabolic flux of $^{13}C_4$, $^{15}N$-aspartate in the biosynthesis of aspartate, glutamate, dihydroorotate, and orotate

The level of phosphorylated CAD was increase in MCF-7 cells but decreased in HeLa cells under hypoxia (Fig. 6a). These data suggest that the hyper-biosynthesis of dihydroorotate most likely is not mediated by hypoxia-regulated protein levels.

Hypoxia also disabled the mitochondrial electron transport chain (ETC) and induced the accumulation of electrons, such as NADH (Fig. 6b). Interestingly, mitochondrial dysfunction was previously reported to promote cells to use glutamine-carbon for acetyl-CoA through the reductive pathway[25]. Here, we confirmed that the inhibition of the ETC by antimycin A-induced NADH accumulation (Fig. 6b) and push glutamine-carbon into acetyl-CoA in HeLa, MCF-7, and 4T1 cells (Supplementary Fig. 14a), but it did not induce HIF-1α (Fig. 6c). We then used HeLa and 4T1 cells to perform a targeted metabolomic analysis. Eighteen overlapped nitrogen-contained metabolites in both HeLa and 4T1 cells were significantly affected by antimycin A (Fig. 6d and Supplementary Fig. 14b), among which eight overlapped

metabolites, including increased dihydroorotate and decreased aspartate and UTP, were also found to be significantly changed by hypoxia. In fact, the level of cellular aspartate was also previously observed to reduce in cells with the ETC dysfunction[26,27]. The tracing analysis showed that antimycin A, similar to hypoxia, also promoted metabolic flux of $^{13}C_5$-glutamine to aspartate $m + 3$ and dihydroorotate $m + 3$ through the reductive pathway (Supplementary Fig. 14c–14e). Importantly, the excretion of dihydroorotate, not orotate, was detected in the culture medium with antimcyin A treatment (Fig. 6e).

Now, we tried to alleviate the electron accumulation in HeLa cells under hypoxia using a pyruvate analog, α-ketobutyrate that can be reduced to excretory α-hydroxybutyrate by NADH-consuming lactate dehydrogenases and thus neutralize NADH accumulation[26]. Our results showed that supplementation of α-ketobutyrate indeed decreased NADH/NAD$^+$ ratio (Fig. 6f). Meantime, we also observed that α-ketobutyrate attenuated the

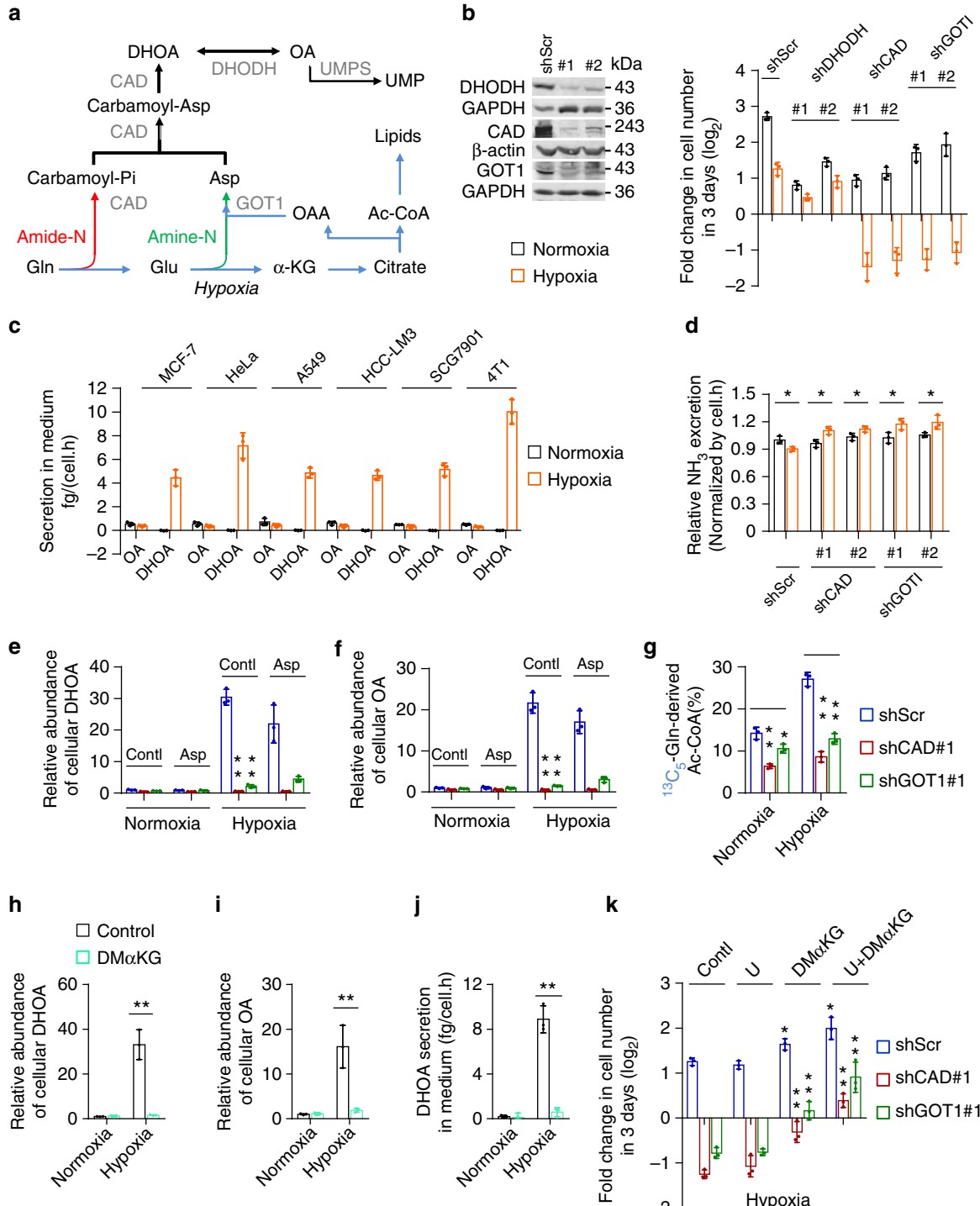

**Fig. 5** Association of glutamine-carbon metabolism with its nitrogen assimilation under hypoxia. **a** A schematic to show the metabolism of glutamine carbon and nitrogen in the pyrimidine biosynthesis. **b** Proliferation of HeLa cells with or without knockdown of DHODH, CAD, and GOT1 cultured under hypoxia and normoxia for 3 days. Values are the means ± SEM of triplicate experiments. Western blot to validate the knockdown of DHODH, CAD, and GOT1. **c** Excretion of dihydroorotate and orotate to medium from cells cultured under hypoxia and normoxia for 8 h. **d** Relative ammonia excretion from HeLa/shScramble, HeLa/shCAD, and HeLa/shGOT1 cells cultured under hypoxia and normoxia for 8 h. Values are the means ± SEM of triplicate experiments. **e**, **f** Relative cellular dihydroorotate and orotate in HeLa/shScramble, HeLa/shCAD, and HeLa/shGOT1 cells cultured under hypoxia and normoxia for 8 h in the presence or absence of 10 mM aspartate. **g** Mass isotopomer analysis of acetyl-CoA in HeLa/shScramble, HeLa/shCAD, and HeLa/shGOT1 cells cultured with the medium containing 1 mM of $^{13}C_5$-glutamine under hypoxia and normoxia for 8 h. **h**, **i** The relative abundance of cellular dihydroorotate and orotate in HeLa cells cultured under hypoxia and normoxia for 8 h in the presence or absence of 2 mM dimethyl α-ketoglutarate (DMαKG). **j** Excretion of dihydroorotate to medium from HeLa cells cultured under hypoxia and normoxia for 8 h in the presence or absence of 2 mM DMαKG. **k** Proliferation of HeLa/shScramble, HeLa/shCAD, and HeLa/shGOT1 cells cultured under hypoxia for 3 days in the presence or absence of 100 μM uridine and/or 2 mM DMαKG. Values are the means ± SEM of triplicate experiments. All cultures were supplied with 10% dialyzed serum. Values are the means ± SEM of three independent experiments, if not specified. *$p < 0.05$; **$p < 0.01$ (Student's $t$-test)

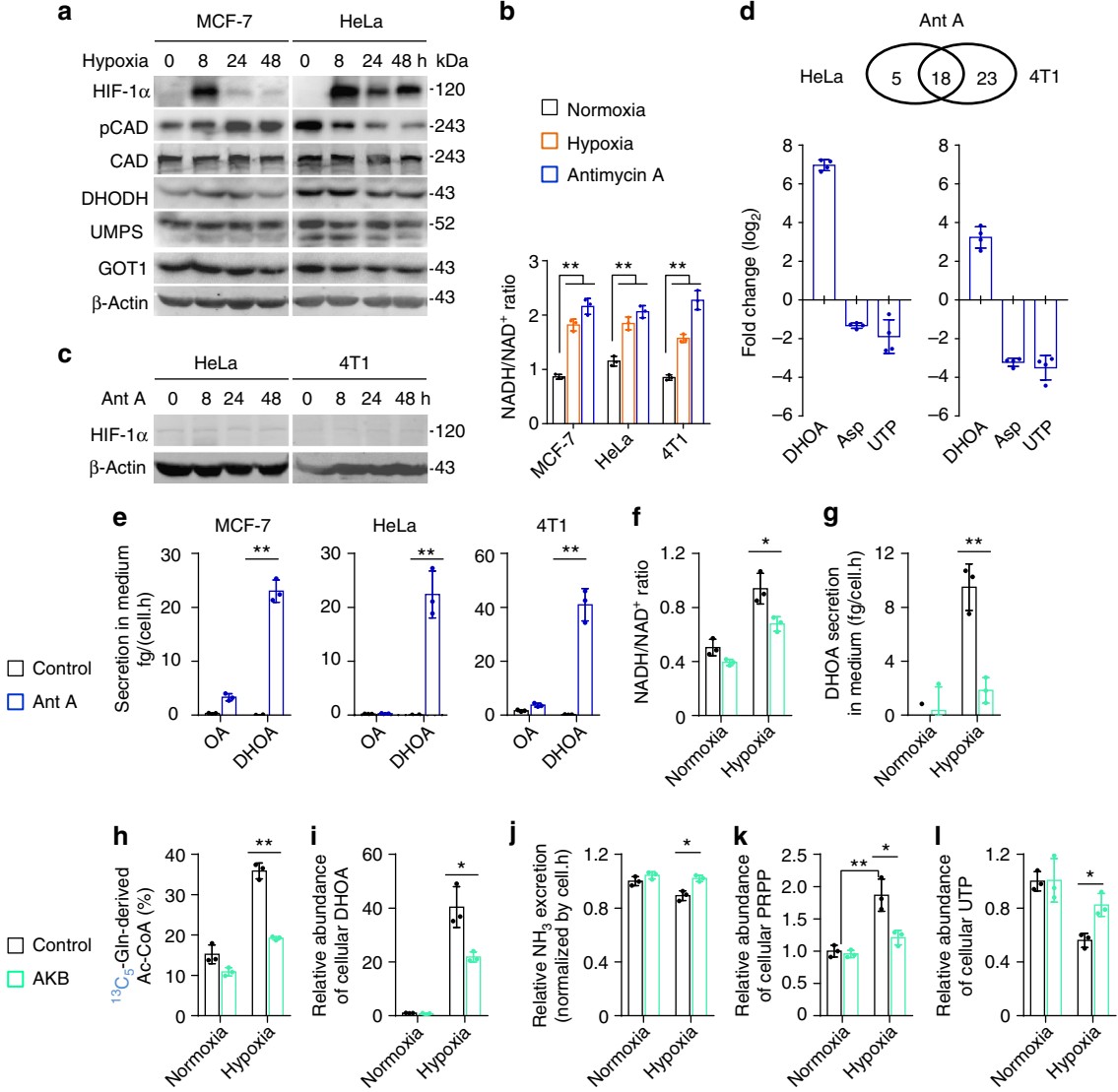

**Fig. 6** Hypoxia-induced NADH accumulation promotes biosynthesis and excretion of dihydroorotate. **a** Western blot of lysates from MCF-7 and HeLa cells cultured under hypoxia for different time as indicated. **b** NADH/NAD$^+$ ratio in MCF-7, HeLa, and 4T1 cells cultured under hypoxia and normoxia for 2 h. Values are the means ± SEM of triplicate experiments. **c** Western blot of lysates from MCF-7 and 4T1 cells cultured under hypoxia for different time as indicated. **d** Targeted metabolomics of HeLa and 4T1 cells cultured under hypoxia or treated with antimycin A (1 μM) for 8 h. The relative abundance of dihydroorotate, asparatate, and UTP were listed here. Values are the means ± SEM of four independent experiments. **e** Excretion of dihydroorotate and orotate to medium from cells treated with or without antimycin A for 8 h. **f** NADH/NAD$^+$ ratio in HeLa cells cultured under hypoxia and normoxia for 2 h in the presence or absence of 1 mM α-ketobutyrate. **g** Excretion of dihydroorotate to medium from HeLa cells cultured under hypoxia and normoxia for 8 h in the presence or absence of 1 mM α-ketobutyrate. **h** Mass isotopomer analysis of acetyl-CoA in HeLa cells cultured with the medium containing 1 mM of $^{13}C_5$-glutamine under hypoxia and normoxia for 8 h in the presence or absence of 1 mM α-ketobutyrate. **i** The relative abundance of cellular dihydroorotate in HeLa cells cultured under hypoxia and normoxia for 8 h in the presence or absence of 1 mM α-ketobutyrate. **j** Relative ammonia excretion from HeLa cells cultured under hypoxia and normoxia for 8 h in the presence or absence of 1 mM α-ketobutyrate. Values are the means ± SEM of triplicate experiments. **k**, **l** The relative abundance of cellular PRPP and UMP in HeLa cells cultured under hypoxia and normoxia for 8 h in the presence or absence of 1 mM α-ketobutyrate. All cultures were supplied with 10% dialyzed serum. Values are the means ± SEM of three independent experiments, if not specified. *$p < 0.05$; **$p < 0.01$ (Student's $t$-test)

excretion of dihydroorotate (Fig. 6g), reduced glutamine-derived acetyl-CoA $m + 2$ (Fig. 6h), decreased the accumulation of cellular dihydroorotate (Fig. 6i) and orotate (Supplementary Fig. 14f), and restored cellular aspartate (Supplementary Fig. 14g) under hypoxia. In addition, α-ketobutyrate supply also enhanced ammonia production under hypoxia (Fig. 6j). Taken together, these data suggest that hypoxia-induced NADH accumulation promotes the biosynthesis and excretion of dihydroorotate, and regulates the metabolism of glutamine-nitrogen.

CAD is a multi-domain enzyme and can be allosterically inhibited by UTP or activated by phosphoribosyl pyrophosphate (PRPP)[28]. In fact, we also detected the increased level of cellular PRPP (Fig. 6k), in addition to the decreased cellular UTP under hypoxia (Fig. 2e). These factors possibly accounted for the promotion of CAD activity by hypoxia. Interestingly, α-ketobutyrate also suppressed PRPP accumulation and reversed cellular UTP under hypoxia (Fig. 6k, l). NADH acts as the coenzyme of many cellular transformations, and thus its accumulation could extensively influence these reactions and

reprogram cellular metabolism. NADH accumulation most likely indirectly activates CAD, given that CAD does not directly employ NADH and NAD$^+$.

**Increased dihydroorotate excretion in in vivo tumors**. In view of the in vivo hypoxic microenvironment of tumors, we then measured the blood dihydroorotate and orotate in patients of breast cancer, lung cancer, gastric cancer, and liver cancer, as well as healthy persons. However, cancer patients' blood orotate, but not dihydroorotate, was detected to significantly enhance compared to the healthy controls (Fig. 7a, b). Furthermore, we measured the levels of blood dihydroorotate and orotate, and both metabolites were found to significantly increase (Fig. 7c, d). To further investigate whether orotate was directly released from tumor or oxidized from dihydroorotate in blood, we administrated mice with intraperitoneal injection with amide-$^{15}$N-labeled glutamine (Supplementary Fig. 15a). Dihydroorotate and orotate in the tumor tissue was rapidly labeled by $^{15}$N (Supplementary Fig. 15b). About 35% and 25% of blood dihydroorotate was labeled by $^{15}$N at 1 and 2 h postinjection in tumor-bearing but not in healthy mice (Fig. 7e and Supplementary 15c, d). In contrast, only <10% of blood orotate was labeled in these mice, and a slightly increased labeled fraction of blood orotate was observed in tumor-bearing mice (Fig. 7f and Supplementary Fig. 15c, d). Similar results were also obtained from 4T1-derived tumor-bearing mice (Supplementary Fig. 15e–15h). These data suggest that the in vivo tumors could directly excrete dihydroorotate that is somehow rapidly converted to orotate in blood (Fig. 7g). The amount of blood orotate was much greater than that of blood dihydroorotate in mice (Fig. 7c, d and

Supplementary Fig. 15e, f), so that the newly synthesized $^{15}$N-labeled orotate derived from dihydroorotate was largely diluted.

Both CAD and DHODH are indispensable to pyrimidine biosynthesis, thus their inhibition by shRNA led to the similarly suppressive effect on cell proliferation in the normal condition (Fig. 5b). It was not surprising that knockdown of CAD and DHODH repressed cell proliferation in a xenograft mouse model (Fig. 7h). However, we found that CAD knockdown suppressed the in vivo tumor growth much more significantly than DHODH knockdown (Fig. 7h). This most likely resulted from the fact that CAD, not DHODH was required for dihydroorotate biosynthesis and cell survival under hypoxia (Fig. 5a, b). Accordingly, higher levels of CAD showed a positive correlation with a shorter overall survival time in patients of breast cancer, lung cancer, gastric cancer, and liver cancer (Supplementary Fig. 16a). In contrast, a high level of DHODH was only observed in patients of gastric cancer with a short overall survival time (Supplementary Fig. 16b). These data suggest that the increased biosynthesis of dihydroorotate from glutamine is critical for the in vivo tumor growth. Compared with CAD, GOT1 seemed to be mainly required for dihydroorotate biosynthesis and cell proliferation under hypoxia but less affected cell growth in the normal condition (Fig. 5b, e), which possibly resulted in a weaker inhibitory effect of GOT1 knockdown on the in vivo cell growth (Fig. 7h).

## Discussion

In this study, we reveal a specific metabolic pathway that hypoxia pushes glutamine carbon and nitrogen via the reductive pathway to dihydroorotate that are then expelled outside of cells rather than processing to their downstream UMP. This unique metabolic reprogramming probably has a vital physiological relevance. Proliferating cancer cells require glutamine carbon to generate

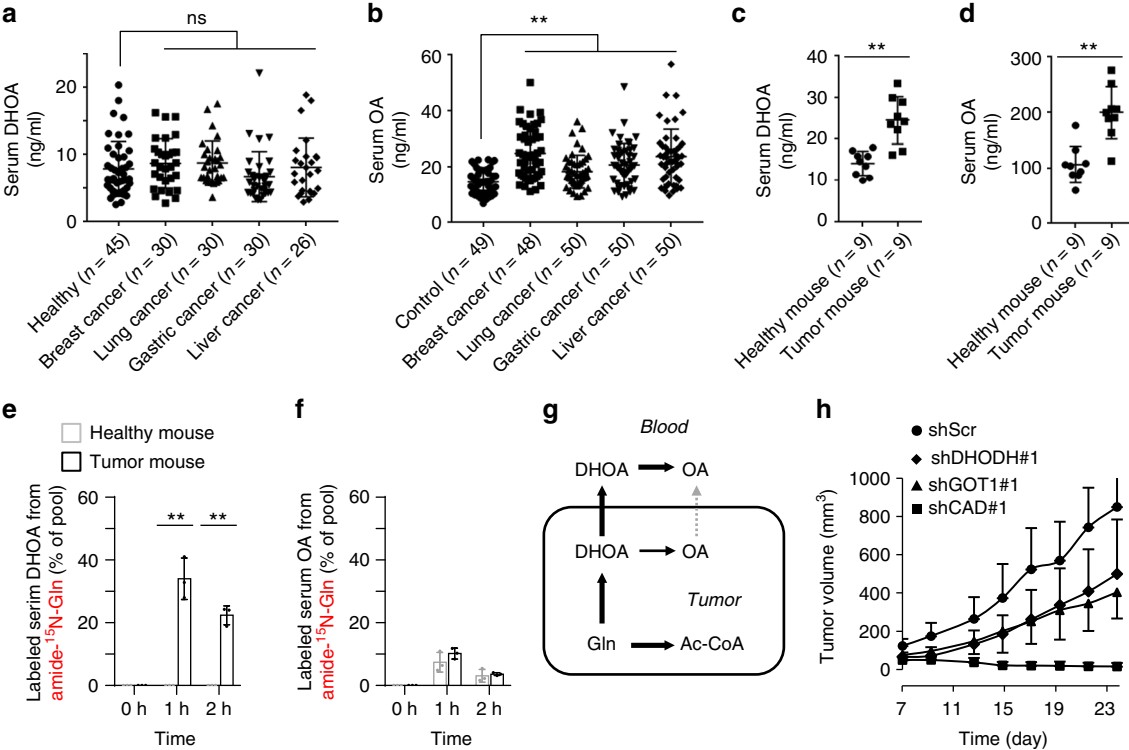

**Fig. 7** Glutamine-derived dihydroorotate is required for tumor growth. **a**, **b** Serum dihydroorotate and orotate in healthy controls and cancer patients. **c**, **d** Serum dihydroorotate and orotate in healthy and HeLa-derived tumor-bearing nude mice. **e**, **f** The $^{15}$N-labeled fraction of blood dihydroorotate and orotate in healthy and HeLa-derived tumor-bearing nude mice intraperitoneally injected with 5 mmol kg$^{-1}$ of amide-$^{15}$N-glutamine for 1 or 2 h. Values are the means ± SEM of data from three mice. **g** Tumors directly excreted dihydroorotate that was oxidized to orotate in blood. **h** The in vivo tumor growth of HeLa/shScramble, HeLa/shDHODH, HeLa/shCAD, and HeLa/shGOT1 cells. *$p < 0.05$; **$p < 0.01$ (Student's $t$-test)

acetyl-CoA for lipid synthesis under hypoxia. During this process, cells need to properly dispose of the concomitant "by-products", nitrogen (ammonia/ammonium), and oxaloacetate that could not be effectively catabolized under hypoxia, from glutamine. The accumulation of these "by-products", particularly ammonia, could be toxic to cells[12,13,21]. The secretion of dihydroorotate perfectly scavenges both the rest nitrogen and carbon and renders glutamine mainly to acetyl-CoA in cells. Hypoxia promotes the enrichment of glutamine-nitrogen in dihydroorotate on one hand, and suppresses the conversion of dihydroorotate to its downstream UMP on the other hand, which could lead to the accumulation of dihydroorotate and promote its excretion.

The amide nitrogen of glutamine is most often used to synthesize asparagine and nucleotide, concomitantly with production of glutamate[29]. Glutamate can be used for multiple purposes, but it can be readily synthesized through the transamination between α-ketoglutarate and other amino acids, thus it is dispensable to cancer cell proliferation. By contrast, the biosynthesis of glutamine from glutamate is inactive in cancer cells. Therefore, the amide-nitrogen of glutamine is necessary to cell growth. Normally, when the amide-nitrogen of glutamine is used, the resultant glutamate, if beyond the metabolic requirement, could be excreted out of cells or replenish the CAC after its transamination. Once the metabolic assimilation of glutamine amide-nitrogen could not keep pace with that of glutamine carbon, cells need to get rid of the superfluous amide-nitrogen. Generally, glutamine is thought to liberate amide-nitrogen as ammonia and converted to glutamate that can further produce α-ketoglutarate upon either deamination or transamination[5,9,11]. The accumulating ammonia should be safely removed. In mammalian cells, there are three enzymes accounting for ammonia assimilation[30,31]. GS synthesizes glutamine from glutamate and ammonia, and this process is essentially a reversion of glutamine deamination and thus does not play a real role in scavenging glutamine-derived ammonia. Carbamoyl phosphate synthetase I (CPSI) incorporates ammonia to urea, but we detected a decreased cellular level of urea in cancer cells under hypoxia (Fig. 1g). GLUD can convert α-ketoglutarate and ammonia to glutamate whose amine group could be further transferred to other amino acids. Unfortunately, GLUD-mediated ammonia assimilation inefficiently or does not take place in cancer cells[4,15], especially in the hypoxic condition (Fig. 3e). Therefore, proliferating cancer cells develop the specific metabolic pathway to dispose of glutamine amide-nitrogen.

In mammalian cells, glutamine amide-nitrogen is used to synthesize carbamoyl phosphate by carbamoylphosphate synthetase II (CPSII) domain of CAD protein, a trifunctional multi-domain enzyme. Carbamoyl phosphate then reacts with aspartate to generate carbamoylaspartate, which is catalyzed by ATCase domain of CAD. The dihydroorotase (DHOase) domain of CAD further synthesizes dihydroorotate from carbamoylaspartate. Although aspartate is the major precursor for pyrimidine biosynthesis, it is seriously scarce in human blood and actually cannot be efficiently absorbed even upon its supplementation[27]. Therefore, the cellular aspartate almost completely depends on its biosynthesis, and its amine-nitrogen is transferred from glutamate. In contrast, glutamine is the most abundant amino acid in human blood, and essentially it can provide directly amide-nitrogen and indirectly amine-nitrogen for nucleotide biosynthesis. Overall, glutamine carbon and nitrogen could be readily coordinatively catabolized without ammonia generation. Under hypoxia, proliferating cells increase the metabolic requirement for glutamine carbon to support lipogensis, and excrete overflowed nitrogen and carbon as the form of dihydroorotate. Moreover, hypoxia-induced NADH accumulation, not the typical hypoxia-associated HIF1 signal pathway, is most likely the cause to drive

the metabolic reprogramming of glutamine. It can enhance the cellular allosteric activator PRPP and reduce the cellular allosteric inhibitor UTP of CAD (Fig. 6k, l), and thus possibly indirectly activate CAD to promote biosynthesis of dihydroorotate from glutamine-derived aspartate. This could provide a complementary explanation for the reduced cellular aspartate in the condition of ETC dysfunction, as previously reported[26,27,32,33].

Dihydroorotate is somehow rapidly converted to orotate in the blood, as suggested in our current study (Fig. 7g). As a fact, urea cycle disorders can also lead to orotic aciduria, a symptom of an increased urinary orotate[34,35], supporting that the excretion of dihydoorotate/orotate is an alternative pathway to dispose of ammonia. Our in vitro data showed that knockdown of CAD enhanced ammonia excretion from cancer cells under hypoxia. Although the in vivo tumor cells often grow in a hypoxic microenvironment, whether their ammonia production will be affected by CAD knockdown remains unclear due to the inefficient tumor formation of cancer cells with shCAD. However, we indeed measure the increased level of blood dihydroorotate/orotate in cancer patients or tumor-bearing mice. It suggests that the specific metabolic flux of glutamine to excretory dihydroorotate/orotate exists in vivo. Therefore, the blood dihydroorotate/orotate may be a potential biomarker for cancer diagnosis, and the involved metabolic enzymes, such as CAD and GOT1, could be potential targets for cancer treatments.

## Methods

**Cell culture.** MCF-7, A549, HeLa, HCC-LM3, SGC-7901, and 4T1 cells were obtained from ATCC. All the cells were maintained in high glucose DMEM supplemented with 10% fetal bovine serum (BioInd, Israel) and 50 IU penicillin/streptomycin (Invitrogen, USA) in a humidified atmosphere with 5% $CO_2$ at 37 °C. Hypoxia studies were carried out at 1% oxygen.

**Cell imaging and cell death assay.** $10 \times 10^4$ cells expressing GC3AI were grown in 12-well plates, and after the desired treatments, cells were then rapidly imaged with an EVOS® FL digital inverted fluorescence microscope with ×10 objective lens. The filter sets for imaging were: GFP: Ex470/22, Em525/50. Cell death assay was performed as previous[36,37], GFP-positive cells were counted as apoptosis. Five random areas in each well were imaged, and each area contained more than one hundred of cells. Cell counting was performed with ImageJ software (1.47) by analyzing these pictures.

**Proliferation assay.** Cells were plated in triplicate in 12-well plates at $5 \times 10^4$ cells per well in 1.5 ml. After days as indicated in experiments, wells were washed twice with PBS buffer to remove dead cells, and then the entire contents of the well were trypsinized. Cell number was determined using a hemocytometer. For each well, the fold change in cell number relative to $Day_0$ was presented in a $log_2$ scale.

**NADH/NAD+ ratio assay.** NADH/NAD+ ratio was done using the NAD/NADH-Glo™ Assay (Cat#G9072 from Promega, USA) according to the manual instructions with modifications. Briefly, $10^5$ cells per well were seeded in 12-well plates for 24 h, and then were incubated with the treatment medium for the desired period. Cells were quickly washed once with PBS, extracted with ice cold lysis buffer (1% dodecyltrimethylammonium bromide (DTAB) in 0.2 N NaOH diluted 1:1 with PBS), immediately frozen at −80 °C and thawed at room temperature. To measure NADH, 150 μl of the samples was moved to wells of 48-well plates and incubated at 60 °C for 15 min, where basic conditions selectively degrade NAD+. To measure NAD+, another 150 μl of the samples was moved to wells of 48-well plates containing 75 μl of 0.4 N HCl and incubated at 60 °C for 15 min, and the acidic conditions selectively degrade NADH. Following incubations, samples were allowed to equilibrate to room temperature and then quenched by neutralizing with 150 μl of HCl/Trizma solution (0.25 M Tris in 0.2 N HCl) (NADH measurement) or 75 μl of 0.5 M Tris base (NAD+ measurement). 50 μl of the neutralized samples was moved to wells of 96-well white luminometer plates (Cat#3925 from CoStar, USA), and mixed with 50 μl of the newly prepared NAD/NADH-Glo™ Detection Reagent. The mixtures were gently shaken for 30 min at room temperature, and the luminescence was measured using a Synergy H1 Hybrid Multi-Mode reader (BioTek, USA).

**Metabolite profiling and isotope tracing.** LC/MS analyses were conducted on a TSQ Quantiva triple quadrupole mass spectrometer networked to a Dionex Ulti-Mate 3000 UPLC system (Thermo Fisher Scientific) at the Metabolomics Facility at Tsinghua University Branch of China National Center for Protein Sciences

(Beijing, China). MRM mode was developed using chemical standards. Experiments were performed in medium containing 10% dialyzed FBS (Gibco). Medium was prepared to contain 100% of either the glucose, glutamine, or aspartate pool labeled with $^{13}$C or $^{15}$N and unlabeled other pool. DMEM lacking glucose, glutamine, and pyruvate was prepared from powder (Sigma), and then supplemented with labeled-glucose or labeled-glutamine, as indicated in the experiments. All the reconstituted experimental media finally contained 10 mM glucose, 1 mM glutamine, and/or 10 mM aspartate if used. Cells were grown in 60-mm dishes until 80% confluent, then rinsed with PBS and cultured with 2 ml isotopes-containing medium for 8 h in the conditions as indicated in the experiments. Cells were then extracted by freeze-thawing three times in 0.5 ml 80% methanol (pre-chilled to −80 °C). Macromolecules and debris were removed by centrifugation at 14,000×g for 20 min at 4–8 °C, and the metabolite-containing supernatants were dried under nitrogen gas. Dried samples were stored at −80 °C and then resuspended in 50 μl 80% methanol and prepared for LC/MS analyses. 1 μl of each sample was injected onto a Synergi Hydro-RP 100A 2.1 × 100 mm column (Phenomenex) for metabolite separation with column temperature at 35 °C. Mobile phases A and B were 10 mM thiobarbituric acid in aqueous with pH 5 and 100% methanol, respectively. The chromatographic gradient was set for mobile phase B as follows: 0–3.5 min: 1% B; 3.5–22 min: from 1% to 70% B; 22–23 min: from 70% to 90% B; 23–25 min: 90% B; 25–30 min: 1% B. Data were acquired using positive/negative switching method. Spray voltages of 3.5 and 2.5 kV were applied for positive and negative modes, respectively. Q1 and Q3 resolution was set at 0.7 and 1 s of cycle time was used in the method. MRM data were analyzed using Tracefinder (Thermo Fisher Scientific) to quantify metabolites for flux analysis. Retention times and mass fragmentation signatures of all metabolites were validated using pure standards. Ion pairs with various isotope labels were derived based on precursors' and fragments' chemical structures. The abundance of each mass isotopomer was then mathematically corrected to eliminate natural abundance isotopes and finally converted into a percentage of the total pool.

For metabolite profiling experiments, cells were grown in 60-mm dishes until 80% confluent, then rinsed with PBS and cultured with 2 ml DMEM containing 10% dialyzed FBS for 8 h under normoxia or hypoxia or in the presence of 1 μM antimycin A. To determine the relative abundance of intracellular metabolites across samples, cells was extracted with 0.5 ml 80% pre-chilled methanol. Samples were prepared and analyzed as described in the above. The areas of the ion peaks of interest were corrected by cell number. Finally, the relative abundance of metabolites was compared with each other.

**Targeted metabolomics**. HeLa and 4T1 cells were grown in 60-mm dishes until 80% confluent, then cultured under hypoxia (1% O$_2$) or treated with antimycin A (1 μM) for 8 h. Samples were prepared as described in the section of "Metabolite profiling and isotope tracing". Samples were randomized, in order to avoid machine drift, and were blinded to the operator.

Analysis was performed on TSQ Quantiva Triple Quadrupole mass spectrometer (Thermo, CA) with positive/negative ion switching at the Metabolomics Facility at Tsinghua University Branch of China National Center for Protein Sciences (Beijing, China). Mobile phase A is prepared by adding 2.376 ml tributylamine and 0.858 ml acetic acid to HPLC-grade water, then adding HPLC-grade water to 1 l volume. Mobile phase B is HPLC-grade methanol. Synergi Hydro-RP 100A column is used for targeted metabolites separation with column temperature at 35 °C. The detailed mass spectrometer parameters are shown as follows: spray voltage, 3.5 and 2.54 kV; sheath gas flow rate, 35 Arb; aux gas flow rate, 12 Arb; ion transfer tube temperature, 320 °C; Q1 and Q3 resolution, 0.7 FWHM; and cycle time, 1 s. MRM data were analyzed using Tracefinder (Thermo Fisher Scientific) to quantify metabolites.

Targeted metabolomics contained 340 ion transitions which were tuned using chemical standards. This method focused on the central carbon metabolism including glycolysis, CAC, purine and pyrimidine metabolism, amino acid metabolism, and related metabolites.

**Metabolite uptake and excretion**. Levels of analine, glutamate, glutamine, orotate, and dihydroorootate were determined using LC/MS. Conditioned medium was sampled after 8 h. 100 μl medium was extracted with 400 μl 100% pre-chilled methanol with 10 ng ml$^{-1}$ $^{15}$N$_2$-orotate as the internal standard. The mixture was centrifuged at 14,000×g for 20 min at 4–8 °C, and the metabolite-containing supernatants were dried under nitrogen gas. Dried samples were stored at −80 °C and then resuspended in 50 μl 80% methanol and prepared for LC/MS analyses as described in the section of "Metabolite profiling and isotope tracing". The amount of orotate can be directly calculated by comparing the area of the ion peak of orotate to that of internal $^{15}$N$_2$-orotate. Since no available dihydroorotate isotope was used as the internal standard, medium dihydroorotate could not be directly quantified. Therefore, 100 μl aliquots of control medium containing 0, 3.125, 6.25, 12.5, 25, 50, 100, and 200 ng ml$^{-1}$ dihydroorotate were in parallel prepared. The areas of the ion peaks of dihydroorotate were corrected by those of the internal $^{15}$N$_2$-orotate. The amount of dihydroorotate can be calculated based on the standard curve. The ion counts of medium glutamine corrected by internal $^{15}$N$_2$-orotate were compared with each other, and the data were presented as the relative uptake. The areas of the ion peaks of analine, glutamate, and glutamine were corrected by those of the internal $^{15}$N$_2$-orotate. The corrected ion peak area was

used to represent the amount of metabolite and data were presented as the relative uptake or excretion to the control.

Ammonia in the medium was determined using the ammonia slides and the VITROS Chemistry Products Calibrator Kit 5 on an autoanalyzer (VITROS 5600 Integrated System, Ortho-Clinical Diagnostics, United States). Briefly, a drop of medium sample was deposited on the slide and evenly distributed by the spreading layer to the underlying layers. Water and nonproteinaceous components travel to the underlying buffered reagent layer, and the ammonium ions are converted to gaseous ammonia. The semi-permeable membrane allows only ammonia to pass through and prevents buffer or hydroxyl ions from reaching the indicator layer. After a fixed incubation period, the reflection density of the dye is measured using the white background of the spreading layer as a diffuse reflector.

Urea in the medium was measured using the autoanalyzer AU5800 (Beckman Coulter, United States). This urea procedure is based on an adaptation of the enzymatic method of Talke and Schubert. In this method, urea is hydrolyzed enzymatically by urease to yield ammonia and carbon dioxide. The ammonia and α-ketoglutaric acid are converted to glutamate in a reaction catalyzed by L-GLUD. Simultaneously, a molar equivalent of reduced NADH is oxidized. Two molecules of NADH are oxidized for each molecule of urea hydrolyzed. The rate of change in absorbance at 340 nm, due to the disappearance of NADH, is directly proportional to the urea concentration in the sample.

Aliquots of the medium without cells under the same conditions were used to measure the concentration of metabolites as the background control. The increased or reduced amount of metabolite in the medium, normalized for area under the curve, was the excretion or uptake of metabolite by a cell per hour.

The cell growth curve is fitted to an exponential equation:

$$y = f(t) = N_{initial} \cdot 2^{t/t_d}$$

where $N$ is cell number, $t$ is time in hours, and $t_d$ is the doubling time for cell proliferation. The area ($S$) under the curve can be obtained by an integral equation:

$$S = \int_0^{t_1} f(t) \cdot dt = N_{initial} \cdot t_1 \cdot \frac{(e^{\ln(N_{final}/N_{initial})} - 1)}{\ln(N_{final}/N_{initial})}$$

where $N_{initial}$ and $N_{final}$ are the initial and final cell numbers that can be experimentally determined, $t_1$ is the treatment time. If $N_{initial} = N_{final}$, $S$ is calculated using the following equation:

$$S = N_{initial} \cdot t_1$$

The metabolite consumption/excretion per cell per hour ($R$) is calculated by the following equation:

$$R = ([Metabolite]_{final} - [Metabolite]_{initial})/S$$

The unit for $R$ is mol per (cell h) or mol per cell per hr.

**Serum metabolite assay**. Blood samples were obtained from cancer patients with pathologic diagnosis at Tianjin Medical University Cancer Institute and Hospital. Informed consent was obtained from all patients according to the regulation of the Institutional Review Boards of Tianjin Medical University Cancer Institute and Hospital in agreement with Declaration of Helsinki.

Hundred microliters of serum was mixed with 400 μl 100% pre-chilled methanol with 10 ng ml$^{-1}$ $^{15}$N$_2$-orotate as the internal standard. After being kept at −80 °C for 2 h, the mixture was centrifuged at 14,000×g for 20 min at 4–8 °C. Four hundred microliters of the metabolite-containing supernatant was transferred to 1.5 EP tube and dried under vacuum conditions. Dried samples were stored at −80 °C and then resuspended in 50 μl 80% methanol. Two microliters of the sample was prepared for LC/MS analyses as described in the section of "Metabolite profiling and isotope tracing". The amount of orotate can be directly calculated by comparing the area of the ion peak of orotate to that of internal $^{15}$N$_2$-orotate. In addition, 100 μl aliquots of water containing 0, 3.125, 6.25, 12.5, 25, 50, 100, or 200 ng ml$^{-1}$ dihydroorotate were prepared, like serum samples, as the standard curve. The areas of the ion peaks of dihydroorotate were corrected by those of the internal $^{15}$N$_2$-orotate. The amount of dihydroorotate can be calculated based on the standard curve.

**Animal experiments**. The animal protocol was approved by the Institute Animal Care and Use Committee at Tianjin Medical University, in accordance with the principles and procedures outlined in the NIH Guide for the Care and Use of Laboratory Animals. Female nude mice (4–5 weeks, 19–20 g) were purchased from the Experimental Animal Center of Nanjing Biomedical Research Institute at Nanjing University. Mice were administered with a standard, housed and maintained in pathogen-free house in a 12:12 h light–dark cycle. Temperature and humidity were maintained at 24 ± 2 °C and 50 ± 5%, respectively.

$5 \times 10^6$ cells of HeLa/shScramble, HeLa/shDHODH#1, HeLa/shCAD#1, or HeLa/shGOT1#1 were injected subcutaneously into the groins of 5-week-old mice, and five mice were used for each cell line. After establishment of palpable tumors,

tumor growth was measured every 3 days using digital calipers, and volumes were calculated using the formula $1/2 \times L \times W^2$. All of the mice were killed at the end and tumors were harvested and weighed.

For in vivo tracing studies, when HeLa/wt-derived tumors reached about 500 mm³, six tumor-bearing mice, as well as six healthy mice, were injected with amide-¹⁵N-glutamine (5.0 mmoles kg⁻¹) in 0.9% NaCl. Mice were sacrificed 2 and 4 h post-injection. A control mouse was injected with an equivalent amount of unlabeled glutamine and sacrificed 2 h after injection. Tumors were excised, flash-frozen, and powderized. Polar metabolites were extracted from 0.1 g tissue in 1 ml 80% MeOH and profiled on LC/MS as described in the section of "Metabolite profiling and isotope tracing". Blood of each mouse was collected from eyeball into 1.5 ml EP tubes and centrifuged at 3000 rpm 10 min to separate plasma. Metabolites were extracted from plasma in 2:8 (v: v) plasma:MeOH and profiled on LC/MS as described in the section of "Serum metabolite assay".

**Western blot**. After desired treatments as specified as indicated, cells were washed twice with PBS and lysed in buffer (20 mM Tris–HCl, pH 7.5, 150 mM NaCl, 1 mM EDTA, 1% Triton X-100, 2.5 mM sodium pyrophosphate, 1 mM β-glycer-opho-sphate, 1 mM sodium vanadate, 1 mg ml⁻¹ leupeptin, 1 mM phenylmethyl-sulfo-nylfluoride). Equal amounts of protein (30 μg) were loaded onto 10% SDS–PAGE gels. Western detection was carried out using a Li-Cor Odyssey image reader (Li-Cor, USA) (Supplementary Fig. 17). The goat anti-rabbit IgG (Cat#C30502-01) and goat anti-mouse IgG (Cat#C30509-01) secondary antibodies were obtained from Li-Cor (USA). The final concentration of the secondary antibodies used was 0.1 μg ml⁻¹ (1:10000 dilution). The primary antibodies against β-Actin (Cat#60008-1, 1:5000 dilution), GOT1 (Cat#14886-1-AP, 1:1000 dilution), DHODH (Cat#14877-1-AP, 1:1000 dilution), HIF1α (Cat#20960-1-AP, 1:1000), and UMPS (Cat#14830-1-AP, 1:1000) were purchased from Proteintech (USA). Antibodies against CAD (Cat#sc-376072 from Santa Cruz, USA) and pCAD (Ser1859) (Cat#70307 from Cell Signaling Technology) were used with a dilution of 1:1000.

**Gene construction and lentivirus production**. All cDNAs were cloned into len-tiviral expression vectors, pCDH-puro-CMV or pCDH-Neo-CMV. The pLKO.1 lentiviral RNAi expression system was used to construct lentiviral shRNA for genes. The sequences of shRNA used in this study included the following: shScramble: CCTAAGGTTAAGTCGCCCTCG, shDHODH-#1: GTGA GAGTTCTGGGCCATAAA, shDHODH-#2: CGATGGGCTGATTGTTACGAA, shGOT1-#1: GCGTTGGTACAATGGAACAAA, shGOT1-#2: GCTAATGACAA TAGCCTAAAT, shCAD-#1: CGAATCCAGAAGGAACGATTT, and shCAD-#2: GCTCCGAAAGATGGGATATAA. Viral packaging was done according to a previously described protocol. Briefly, expression plasmids pCDH-CMV-cDNA, pCMV-dR8.91, and pCMV-VSV-G were co-transfected into 293T cells using the calcium phosphate coprecipitation at 20:10:10 μg (for a 10-cm dish). The trans-fection medium containing calcium phosphate and plasmid mixture was replaced with fresh complete medium after incubation for 5 h. Media containing virus was collected 48 h after transfection and then concentrated using Virus Concentrator Kit (Biophay, China). The virus was resuspended in the appropriate amount of complete growth medium and stored at −80 °C. Cancer cells were infected with the viruses at the titer of 100% infection in the presence of polybrene (10 μg ml⁻¹) for 48 h, and then cells were selected with puromycin or neomycin.

**Statistics**. Data are given as means ± SD. Statistical analyses were performed using unpaired, two-tailed Student's $t$-test for comparison between two groups. Asterisks in the figure indicated statistical significances ($^*p < 0.05$; $^{**}p < 0.01$).

**Reporting summary**. Further information on experimental design is available in the Nature Research Reporting Summary linked to this Article.

## Data availability
The data that support the findings of this study are available within the Article and its Supplementary Information or from the corresponding author upon reasonable request.

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

## Acknowledgements

We thank Dr. Ceshi Chen (Kunming Institute of Zoology, Chinese Academy of Sciences), Dr. Shimin Zhao (Fudan University), and Dr. Wei Du (University of Chicago, USA) for thoughtful discussion, and also thank Dr. Xiaohui Liu at the Metabolomics Facility at Tsinghua University Branch of China National Center for Protein Sciences (Beijing, China) for technical help. This work was supported by Grants 81622037 and 81672762 from Natural Science Foundation of China, and Start Grant from Advanced Innovation Center for Human Brain Protection.

## Author contributions

Y.W., C.B., and Y.R. performed experiments to determine cell proliferation, cell death, and metabolite quantifications. Y.W., C.B., and M.L. prepared samples for LCMS quantification. Y.R., Q.C., and L.Q. created cell lines and performed some proliferation rate experiments. C.Y. provided some ideas. B.L. designed the study and wrote the manuscript.

## Additional information

**Competing interests:** The authors declare no competing interests.

