## [Peer Review File · Nature Communications]

Reviewers' Comments:

Reviewer #1:

Remarks to the Author:

In this manuscript the authors examine glutamine metabolism during normoxia and hypoxia and determine that ammonia utilization changes upon hypoxia – away from GLUD1 metabolism, and toward CAD to excrete/detoxify ammonia. Overall, this is an interesting study, but it is too preliminary to warrant the current conclusions. There are several inconsistencies in the study and controls/experiments are lacking. Specific concerns are listed below.

1. Figure 1 does not optimally set the stage for the study. As the main focus of the paper is glutamine metabolism and proliferation in hypoxia, showing proliferation and metabolism of cancer cells +/- glutamine in normoxia is not important for the story and can be moved to the supplementary.
2. They show that glutamate (2-10mM) doesn't not rescue proliferation of cells under glutamine deprivation due to low levels of GS. Upon GS overexpression, addition of 10mM Glutamate rescued proliferation under glutamine deprivation. 10mM Glutamate seems excessive, and it is surprising that 2mM glutamate doesn't work even under GS overexpression.
3. In figure 1d, 1e, 1g, 1h, it is technically appropriate to show the fractional labeling rather than the single mass isotopomers and then m+n...
4. In Figure 2c - why is glutamate excreted and seems to decrease under hypoxia? This should be explained. The Christian M. Metallo study cited shows that under hypoxia, the glutamine flux to glutamate is increased under hypoxia (figure 3C) instead.
5. Overall: since they are claiming that the NH₃ excretion in the media is decreased under hypoxia, the authors need a control sample lacking cells. Is degradation of glutamine in the media slowed under hypoxia?
6. In the heatmap in Figure 3, are these analyses comparing normoxia and hypoxia? It seems the authors examined hypoxia, but then the comparison is unclear. Also, it is not clear why the cell system has shifted to 4T1 mouse cells. In this case, the above experiments should be performed also with these cells.
7. Figures 5A and 5B are inconsistent with their finding in Figure 2F, where the authors show that glutamine flux to glutamate is increased under hypoxia.
8. A critical experiment to test the authors hypothesis that glutamine flux to DHOA is increased under hypoxia to prevent 'toxic ammonia buildup' is to knockdown CAD enzymes and measure flux during hypoxia, as well as 'ammonia toxicity'.
9. In Figure 6, the metabolic regulation (cofactor) versus enzyme expression is an interesting model. However, more experiments are needed to confirm this model. Does alteration of NADH/NAD⁺ ratio influence CAD activity and ammonia accumulation?
10. Further in vivo studies, using CAD KD models, are required to test whether CAD affects ammonia accumulation or glutamine metabolism in the tumor in vivo.

Reviewer #2:

Remarks to the Author:

The authors address the question of how cells under hypoxia cope with excess nitrogen produced as a consequence of the increased use of glutamine for lipogenesis. They find that DHOA production is greatly increased in the hypoxic cells and that much of this is excreted. They conclude that DHOA is the vehicle for disposal of excess nitrogen. This is an interesting idea but the authors seem to fall short of demonstrating such a role for DHOA production

1. The authors show, as previously reported, that cells under hypoxia take up more glutamine and also show evidence of ETC dysfunction, with increased NADH driving reductive reactions.
2. As expected, ETC dysfunction results in less aspartate production, although the authors seem to interpret this reduction in aspartate levels as evidence of increased flux into DHOA. I don't think

this has been convincingly demonstrated.

3. The increase in DHOA levels could be the result of accumulation due to high NADH levels (which would also affect DHODH reaction to favour accumulation of DHOA) and lack of aspartate to complete the pyrimidine synthesis reactions. I don't think the data distinguish between increased DHOA production and reduction in DHOA utilisation (leading to accumulation)

4. In Figure 3 show increase in OA too –although it is not clear why this happens. Is this also excreted?

5. Does GOT1 depletion prevent DHOA production and an increase in intracellular ammonia? Can this be rescued by feeding the cells high levels of exogenous aspartate? Flux of labelled aspartate could be traced.

6. What is the effect of CAD depletion on glutamine metabolism?

7. Are the effects of CAD depletion rescued by providing cells with nucleotides under normoxia and hypoxia? If DHOA production is critical under hypoxia, nucleotides might only help in normoxic conditions.

Reviewers' comments:

Reviewer #1 (Remarks to the Author):

In this manuscript the authors examine glutamine metabolism during normoxia and hypoxia and determine that ammonia utilization changes upon hypoxia – away from GLUD1 metabolism, and toward CAD to excrete/detoxify ammonia. Overall, this is an interesting study, but it is too preliminary to warrant the current conclusions. There are several inconsistencies in the study and controls/experiments are lacking. Specific concerns are listed below.

1. Figure 1 does not optimally set the stage for the study. As the main focus of the paper is glutamine metabolism and proliferation in hypoxia, showing proliferation and metabolism of cancer cells +/- glutamine in normoxia is not important for the story and can be moved to the supplementary.

Response: Thanks for your suggestions. We already moved them to the supplementary data (Extended Data Figure 1 and 2).

2. They show that glutamate (2-10mM) doesn't not rescue proliferation of cells under glutamine deprivation due to low levels of GS. Upon GS overexpression, addition of 10mM Glutamate rescued proliferation under glutamine deprivation. 10mM Glutamate seems excessive, and it is surprising that 2mM glutamate doesn't work even under GS overexpression.

Response: Our results showed that 2 mM glutamate already significantly rescued cell proliferation of MCF-7/GS cells while 10 mM glutamate almost fully restored cell proliferation (Figure 1i in the old version, or Extended Data Figure 1e in the updated version).

3. In figure 1d, 1e, 1g, 1h, it is technically appropriate to show the fractional labeling rather than the single mass isotopomers and then m+n...

Response: Thanks for your suggestions. Yes, it is easier to understand by showing the labeled fractions. We already reorganized these data as Extended Data Figure 2d and 2e in the updated version.

4. In Figure 2c - why is glutamate excreted and seems to decrease under hypoxia? This should be explained. The Christian M. Metallo study cited shows that under hypoxia, the glutamine flux to glutamate is increased under hypoxia (figure 3C) instead.

Response: Thanks. In fact, our results were consistent with theirs. In Christian M. Metallo's paper, Figure 3c was to show relative contribution of reductive glutamine metabolism to TCA metabolites, determined by M1 labeling from [1-¹³C]-glutamine. Similarly, our results (Figure 2f in the old version or Figure 1e and 5 in the updated version) also showed an increased metabolic flux of glutamine-carbon to glutamate. The Christian M. Metallo study showed that "we observed increased glutamine consumption when A549 cells were cultured at approximately 1% oxygen while glutamate secretion remained unchanged (Supplementary Fig. 1a)". In our paper Figure 2C in the old version (or Extended Data Figure 3a in the

updated version) showed a slight decrease in glutamate excretion under hypoxia. The difference might result from different cell lines. The increased uptake of glutamine but decreased or unchanged excretion of glutamate in cells means an elevated intracellular utilization of glutamine-carbon under hypoxia. We explained it in the updated version as the following: “Indeed, we detected a significantly increased uptake of glutamine but decreased excretion of glutamate in MCF-7, HeLa and 4T1 cells (Fig. 1b and Extended Data Fig. 3a), suggesting an elevated intracellular utilization of glutamine-carbon under hypoxia.”.

5. Overall: since they are claiming that the NH₃ excretion in the media is decreased under hypoxia, the authors need a control sample lacking cells. Is degradation of glutamine in the media slowed under hypoxia?

Response: In the standard protocols to measure the uptake and excretion of metabolites from cultured cells, the medium without cells under the same conditions were always used as the background control. To emphasize this, we already added some descriptions in the part of “Metabolite uptake and excretion” in Methods section as “Aliquots of the medium without cells under the same conditions were used to measure the concentration of metabolites as the background control. The increased or reduced amount of metabolite in the medium, normalized for area under the curve, was the excretion or uptake of metabolite by a cell per hour.”

6. In the heatmap in Figure 3, are these analyses comparing normoxia and hypoxia? It seems the authors examined hypoxia, but then the comparison is unclear. Also, it is not clear why the cell system has shifted to 4T1 mouse cells. In this case, the above experiments should be performed also with these cells.

Response: In the heatmap in Figure 3 (Figure 2a in the updated version), the data indicated the fold change (Log₂) of metabolite abundance in cells under hypoxia compared to those under normoxia. The data under normoxia were normalized as 1, and thus usually not shown. Almost all the heatmap data are presented like this.

At the beginning, we used HeLa and 4T1 cells as the models to narrow the altered cellular metabolites by hypoxia (Figure 2a in the updated version) or antimycin A (Extended Data Figure 14b in the updated version) in our project, because the two cells are easy to form tumors in the xenograft model. In the following molecular experiments, we felt that using MCF-7 human breast cancer cells could be better than 4T1 mouse breast cancer cells. As you suggested, we added some more data on 4T1 cells (Figure 1b-1g).

7. Figures 5A and 5B are inconsistent with their finding in Figure 2F, where the authors show that glutamine flux to glutamate is increased under hypoxia.

Response: Figure 2F (Figure 1e in the updated version) showed the labeling of amino acids by glutamine-carbon (¹³C), while Figure 5a and 5b (Figure 3d and 3e in the updated version) were the labeling of amino acids by glutamine-nitrogen (¹⁵N). Glutamine-carbon labels glutamate, proline (Figure 1e in the updated version) and aspartate (Figure 3f in the updated version), but does not label alanine (Extended Data Figure 5 in the updated version). In contrast, glutamine-amine-¹⁵N labels glutamate, proline, aspartate and alanine (Figure 3d in the updated version) while glutamine-amide-¹⁵N predominantly labels asparagine (Figure 3e

in the updated version). These results were not inconsistent with each other, and just revealed that glutamine-carbon and -nitrogen have different metabolic fates. Therefore, the glutamine-carbon and -nitrogen requires coordinative metabolism.

8. A critical experiment to test the authors hypothesis that glutamine flux to DHOA is increased under hypoxia to prevent 'toxic ammonia buildup' is to knockdown CAD enzymes and measure flux during hypoxia, as well as 'ammonia toxicity'.

Response: Thanks for your constructive suggestions. We already carried out the experiments about the effects of CAD knockdown, as well as GOT1 knockdown, on the metabolism of glutamine-carbon and nitrogen (Figure 5d-5g, 5k and Extended Data Figure 12b-12d in the updated version). We added the related descriptions about these results as a part of "Association of glutamine-carbon metabolism with its nitrogen assimilation under hypoxia" in Results section: "The reprogrammed metabolic pathway essentially renders glutamine only to acetyl-CoA for lipogenesis under hypoxia, and glutamine-amide and -amine groups are incorporated into secretory dihydroorotate by CAD and GOT1. These observations could explain why hypoxia increased the utilization of glutamine-carbon but decreased the release of ammonia (Fig. 1). This speculation was further supported by the results that knockdown of CAD or GOT1 enhanced the production of ammonia under hypoxia (Fig. 5d).

As expected, knockdown of CAD or GOT1 dramatically suppressed hypoxia-induced dihydroorotate and orotate (Fig. 5e,f). Moreover, supplementation with aspartate did not effectively restore the accumulation of dihydroorotate and orotate even in cells with GOT1 depletion (Fig. 5e,f), although the absorbed aspartate can be used to synthesize dihydroorotate and orotate under hypoxia (Fig. 4a,b). Meantime, we found that knockdown of CAD or GOT1 also reduced glutamine-derived α -ketoglutarate, citrate and acetyl-CoA (Fig. 5g and Extended Data Fig. 12b,c). These data suggest that glutamine-carbon metabolism is associated with its nitrogen assimilation to dihydroorotate. To further confirm this speculation, we treated HeLa cells with α -ketoglutarate, the carbon form of glutamine. Our results showed that α -ketoglutarate supplementation almost completely scavenged the accumulated cellular dihydroorotate and orotate (Fig. 5h,i), and suppressed the excretion of dihydroorotate (Fig. 5j). Supplementation with uridine and/or α -ketoglutarate increased proliferation rate of HeLa/shCAD and HeLa/shGOT1 cells in the normal condition (Extended Data Fig. 12d), but only α -ketoglutarate substantially restored cell proliferation (Fig. 5k). Taken together, our results suggest that the metabolism of glutamine-carbon is necessary to cell survival and depends on the increased biosynthesis of dihydroorotate under hypoxia."

9. In Figure 6, the metabolic regulation (cofactor) versus enzyme expression is an interesting model. However, more experiments are needed to confirm this model. Does alteration of NADH/NAD⁺ ratio influence CAD activity and ammonia accumulation?

Response: Thanks for your suggestions. We already did the experiments about the NADH/NAD⁺ ratio by α -ketobutyrate supplementation on ammonia production (Figure 6l in the updated version), and added the descriptions of " α -ketobutyrate supply also enhanced ammonia production under hypoxia (Fig. 6j)" in the last paragraph of "Hypoxia-induced electron accumulation promotes biosynthesis and excretion of dihydroorotate" in Results section.

Our results clearly showed that supplementation of α -ketobutyrate decreased NADH/NAD⁺ ratio (Fig. 6f), and reduced the excretion of dihydroorotate (Fig. 6g) and the accumulation of cellular dihydroorotate and orotate (Fig. 6i and Extended Data Fig. 12f) induced by hypoxia. These results suggest that the accumulated NADH promotes CAD-mediated dihydroorotate biosynthesis.

We also added more results (Fig. 6k,l in the updated version) to show that alteration of NADH/NAD⁺ ratio could influence CAD activity indirectly. CAD is a multi-domain enzyme and can be allosterically inhibited by UTP or activated by PRPP (phosphoribosyl pyrophosphate). In fact, we also found the increased level of cellular PRPP (Fig. 6k), in addition to the decreased cellular UTP under hypoxia (Fig. 2e). These factors possibly accounted for the promotion of CAD activity by hypoxia. Interestingly, α -ketobutyrate also suppressed PRPP accumulation and reversed cellular UTP under hypoxia (Fig. 6k,l). NADH acts as the coenzyme of many cellular transformations, and thus its accumulation could extensively influence these reactions and reprogram cellular metabolism. NADH accumulation most likely indirectly activates CAD, given that CAD does not directly employ NADH and NAD⁺. We added these descriptions to the content of “Hypoxia-induced electron accumulation promotes biosynthesis and excretion of dihydroorotate” in the updated version.

10. Further in vivo studies, using CAD KD models, are required to test whether CAD affects ammonia accumulation or glutamine metabolism in the tumor in vivo.

Response: Thanks for your suggestions. shCAD cells do not effectively form tumors *in vivo*, and thus it is very difficult to test its effect of shCAD on ammonia accumulation or glutamine metabolism *in vivo*. Therefore, we did these experiments in the cell model, please see Response to Question#8.

In addition, we re-wrote the contents about animal experiments with shCAD and shDHODH to explain the importance of CAD-mediated dihydroorotate *in vivo*. “Both CAD and DHODH are indispensable to pyrimidine biosynthesis, thus their inhibition by shRNA led to the similarly suppressive effect on cell proliferation in the normal condition (Fig. 5b). It was not surprising that knockdown of CAD and DHODH repressed cell proliferation in a xenograft mouse model (Fig. 7h). However, we found that CAD knockdown suppressed the *in vivo* tumor growth much more significantly than DHODH knockdown (Fig. 7h). This most likely resulted from the fact that CAD, not DHODH was required for dihydroorotate biosynthesis and cell survival under hypoxia (Fig. 5a,b). Accordingly, higher levels of CAD showed a positive correlation with a shorter overall survival time in patients of breast cancer, lung cancer, gastric cancer and liver cancer (Extended Data Fig. 16a). In contrast, a high level of DHODH was only observed in patients of gastric cancer with a short overall survival time (Extended Data Fig. 16b). These data suggest that the increased biosynthesis of dihydroorotate from glutamine is critical for the *in vivo* tumor growth.” We already added these descriptions to the part of “Increased dihydroorotate biosynthesis and excretion in *in vivo* Tumors” in the updated version.

Reviewer #2 (Remarks to the Author):

The authors address the question of how cells under hypoxia cope with excess nitrogen produced as a consequence of the increased use of glutamine for lipogenesis. They find that DHOA production is greatly increased in the hypoxic cells and that much of this is excreted. They conclude that DHOA is the vehicle for disposal of excess nitrogen. This is an interesting idea but the authors seem to fall short of demonstrating such a role for DHOA production

1. The authors show, as previously reported, that cells under hypoxia take up more glutamine and also show evidence of ETC dysfunction, with increased NADH driving reductive reactions.

Response: Thanks for your summary on our manuscript.

2. As expected, ETC dysfunction results in less aspartate production, although the authors seem to interpret this reduction in aspartate levels as evidence of increased flux into DHOA. I don't think this has been convincingly demonstrated.

Response: Sorry for our poor writing to make you misunderstand it. We concluded that aspartate was promoted to DHOA by hypoxia, based on the metabolic flux experiments not on the reduced level of cellular aspartate. Incorporation of glutamine-amide, -amine ^{15}N and ^{13}C to DHOA was significantly increased under hypoxia (Fig. 3b, 3c and 3i in the updated version). Since glutamine-carbon has to be first converted to aspartate, accompanying with glutamine-carbon-derived acetyl-CoA, and then to DHOA and OA (Fig. 3a in the updated version), we concluded "these data suggest that the biosyntheses of acetyl-CoA, dihydroorotate and orotate from glutamine were urged by hypoxia". To help readers to better understand our conclusion, we now put together all the data about the metabolic flux of isotope-labeled glutamine to DHOA as Figure 3 (in the updated version). In addition, we also removed the descriptions of "Given that the cellular level of aspartate significantly decreased while those of the downstream carbamoyl-Asp, dihydroorotate and orotate dramatically increased (Fig. 2d), aspartate is possibly largely promoted by hypoxia to its downstream products" to avoid misunderstanding.

To further demonstrate the increased assimilation of aspartate into DHOA under hypoxia, we treated cells directly with $^{13}\text{C}_4, ^{15}\text{N}$ -labeled aspartate (New data Fig. 4 in the updated version). In the normal condition, aspartate inefficiently labeled dihydroorotate, orotate and UMP (Fig. 4a,b in the updated version). However, hypoxia promoted the labeling of dihydroorotate and orotate, but not UMP, by $^{13}\text{C}_4, ^{15}\text{N}$ -aspartate m+5 and $^{13}\text{C}_4$ -aspartate m+4. In particular, the newly-synthesized $^{13}\text{C}_4$ -aspartate m+4 was more efficiently incorporated into dihydroorotate and orotate than the absorbed $^{13}\text{C}_4, ^{15}\text{N}$ -aspartate m+5. These data support that hypoxia strongly boosts entry of aspartate to dihydroorotate and orotate. Please see more details in the last paragraph of "Promotion of aspartate to pyrimidine precursors under

hypoxia” in Results section.

Although it has been recently reported that ETC dysfunction resulted in a decreased level of cellular aspartate (we cited references), here our observations provided a complementary explanation that hypoxia strongly boosts enter of aspartate to dihydroorotate and orotate, which possibly leads to the decreased cellular aspartate. We discussed it in Discussion section. However, our manuscript focused on the coordinative metabolism of glutamine-carbon and nitrogen, despite some overlaps with aspartate metabolism.

3. The increase in DHOA levels could be the result of accumulation due to high NADH levels (which would also affect DHODH reaction to favour accumulation of DHOA) and lack of aspartate to complete the pyrimidine synthesis reactions. I don't think the data distinguish between increased DHOA production and reduction in DHOA utilisation (leading to accumulation).

Response: Thanks for your suggestion and reminding. I checked through our manuscript again, and it seemed that we did not conclude the accumulation of DHOA resulted from the increased DHOA production. Possibly, the sentence of “Given that the cellular level of aspartate significantly decreased while those of the downstream carbamoyl-Asp, dihydroorotate and orotate dramatically increased, aspartate is possibly largely promoted by hypoxia to its downstream products” in the old version misguided the readers. Now we deleted this sentence. For sure, the accumulation of a metabolite can result from its increased biosynthesis or decreased utilization.

On one hand, our results showed that DHOA and OA accumulated but they did not process to the downstream UMP under hypoxia. Thus, the high NADH level seemed not to affect the DHODH-mediated biosynthesis of OA from DHOA too much under hypoxia. In addition, aspartate was not required for and its lack should not directly block the biosynthesis of pyrimidine from OA. Anyway, these results just demonstrate that hypoxia can suppress the conversion of DHOA and OA to UMP, i.e. DHOA utilization.

On the other hand, we used isotope tracing to detect the activity of biosynthesis of DHOA under hypoxia. The labeling of DHOA by glutamine-amide, -mine ^{15}N and ^{13}C comprehensively enhanced under hypoxia (Fig. 3b, 3c and 3i), indicating the active biosynthesis. Moreover, our new data of tracing with $^{13}\text{C}_4$, ^{15}N -labeled aspartate also showed that hypoxia preferentially promoted DHOA biosynthesis from newly-synthesized aspartate instead of absorbed aspartate (New data Fig. 4 in the updated version).

Therefore, in our paper we actually emphasized both increased DHOA production (most often described as “enrichment of glutamine-nitrogen in DHOA”) and reduction in DHOA utilization. We summarized it as “Hypoxia promotes the enrichment of glutamine-nitrogen in dihydroorotate on one hand, and suppresses the conversion of dihydroorotate to its downstream UMP on the other hand, which could lead to the accumulation of dihydroorotate and promote its excretion” in the first paragraph of Discussion section.

4. In Figure 3 show increase in OA too –although it is not clear why this happens. Is this also excreted?

Response: We did not measure the excretion of OA (Figure 5c in the updated version). Although we do not know the exactly mechanism, we explained it in the content by “The

overall conversion of glutamine and dioxide carbon to acetyl-CoA and secretory dihydroorotate, not orotate, consumes electrons (Extended Data Fig. 13 in the updated version), which could remit hypoxia-induced electron accumulation”.

5. Does GOT1 depletion prevent DHOA production and an increase in intracellular ammonia? Can this be rescued by feeding the cells high levels of exogenous aspartate? Flux of labelled aspartate could be traced.

Response: Thanks for your constructive suggestions. We did the experiments about effects of GOT1 depletion as well as CAD depletion on the production of DHOA and ammonia (New data Fig. 5d, 5e-5g). Knockdown of CAD or GOT1 dramatically suppressed hypoxia-induced dihydroorotate and orotate (Fig. 5e,f). Moreover, supplementation with aspartate did not effectively restore the accumulation of dihydroorotate and orotate even in cells with GOT1 depletion (Fig. 5e,f), although the absorbed aspartate can be used to synthesize dihydroorotate and orotate under hypoxia (Fig. 4a,b). Meantime, we found that knockdown of CAD or GOT1 also reduced glutamine-derived α -ketoglutarate, citrate and acetyl-CoA (Fig. 5g and Extended Data Fig. 12b,c). These data suggest that glutamine-carbon metabolism is associated with its nitrogen assimilation to dihydroorotate. To further confirm this speculation, we treated HeLa cells with α -ketoglutarate, the carbon form of glutamine. Our results showed that α -ketoglutarate supplementation almost completely scavenged the accumulated cellular dihydroorotate and orotate (Fig. 5h,i), and suppressed the excretion of dihydroorotate (Fig. 5j). Supplementation with uridine and/or α -ketoglutarate increased proliferation rate of HeLa/shCAD and HeLa/shGOT1 cells in the normal condition (Extended Data Fig. 12d), but only α -ketoglutarate substantially restored cell proliferation (Fig. 5k). Taken together, our results suggest that the metabolism of glutamine-carbon is necessary to cell survival and depends on the increased biosynthesis of dihydroorotate under hypoxia. We added the related descriptions about these results as a part of “Association of glutamine-carbon metabolism with its nitrogen assimilation under hypoxia” in Results section.

We cultured HeLa and MCF-7 cells with $^{13}\text{C}_4, ^{15}\text{N}$ -labeled aspartate, and traced the isotope-labeled intermediates in the pyrimidine biosynthesis. Both cell lines substantially absorbed exogenous $^{13}\text{C}_4, ^{15}\text{N}$ -labeled aspartate m+5 (Fig. 4a,b), and we also detected the newly-synthesized $^{13}\text{C}_4$ -labeled aspartate m+4, ^{15}N -labeled aspartate m+1 and ^{15}N -labeled glutamate m+1 (Fig. 4a,b). $^{13}\text{C}_4$ -aspartate m+4 carried glutamine-derived amine nitrogen, while ^{15}N -aspartate m+1 contained $^{13}\text{C}_4, ^{15}\text{N}$ -aspartate-derived amine nitrogen mediated by ^{15}N -glutamate m+1 (Fig. 4c). Hypoxia increased the uptake and biosynthesis of aspartate (Fig. 4a,b). Interestingly, in the normal condition, aspartate inefficiently labeled dihydroorotate, orotate and UMP (Fig. 4a,b). However, hypoxia promoted the labeling of dihydroorotate and orotate, but not UMP, by $^{13}\text{C}_4, ^{15}\text{N}$ -aspartate m+5 and $^{13}\text{C}_4$ -aspartate m+4. In particular, the newly-synthesized $^{13}\text{C}_4$ -aspartate m+4 was more efficiently incorporated into dihydroorotate and orotate than the absorbed $^{13}\text{C}_4, ^{15}\text{N}$ -aspartate m+5. Note: all the figure numbers

6. What is the effect of CAD depletion on glutamine metabolism?

Response: Thanks for your constructive suggestions. We did the experiments about effect of CAD depletion, as well as GOT1 depletion, on glutamine metabolism (New data Fig. 5d, 5e-5g, and Extended Data Fig. 12b and 12c). Knockdown of CAD or GOT1 dramatically

suppressed hypoxia-induced dihydroorotate and orotate (Fig. 5e,f). Moreover, supplementation with aspartate did not effectively restore the accumulation of dihydroorotate and orotate even in cells with GOT1 depletion (Fig. 5e,f), although the absorbed aspartate can be used to synthesize dihydroorotate and orotate under hypoxia (Fig. 4a,b). Meantime, we found that knockdown of CAD or GOT1 also reduced glutamine-derived α -ketoglutarate, citrate and acetyl-CoA (Fig. 5g and Extended Data Fig. 12b,c). These data suggest that glutamine-carbon metabolism is associated with its nitrogen assimilation to dihydroorotate. To further confirm this speculation, we treated HeLa cells with α -ketoglutarate, the carbon form of glutamine. Our results showed that α -ketoglutarate supplementation almost completely scavenged the accumulated cellular dihydroorotate and orotate (Fig. 5h,i), and suppressed the excretion of dihydroorotate (Fig. 5j). Supplementation with uridine and/or α -ketoglutarate increased proliferation rate of HeLa/shCAD and HeLa/shGOT1 cells in the normal condition (Extended Data Fig. 12d), but uridine alone did not while α -ketoglutarate alone or in combination with uridine substantially restored cell proliferation under hypoxia (Fig. 5k). Taken together, our results suggest that the metabolism of glutamine-carbon is necessary to cell survival and depends on the increased biosynthesis of dihydroorotate under hypoxia.

7. Are the effects of CAD depletion rescued by providing cells with nucleotides under normoxia and hypoxia? If DHOA production is critical under hypoxia, nucleotides might only help in normoxic conditions.

Response: Thanks for your constructive suggestions. We did the experiments about effects of uridine and/or α -ketoglutarate on cell proliferation (New data Fig. 5k and Extended Data Fig. 12d). Our results showed that α -ketoglutarate supplementation almost completely scavenged the accumulated cellular dihydroorotate and orotate (Fig. 5h,i), and suppressed the excretion of dihydroorotate (Fig. 5j). Supplementation with uridine and/or α -ketoglutarate increased proliferation rate of HeLa/shCAD and HeLa/shGOT1 cells in the normal condition (Extended Data Fig. 12d), but uridine alone did not while α -ketoglutarate alone or in combination with uridine substantially restored cell proliferation under hypoxia (Fig. 5k). Taken together, our results suggest that the metabolism of glutamine-carbon is necessary to cell survival and depends on the increased biosynthesis of dihydroorotate under hypoxia. We added these descriptions in the part of “Association of glutamine-carbon metabolism with its nitrogen assimilation under hypoxia.” in Results section.

Reviewers' Comments:

Reviewer #1:

Remarks to the Author:

The authors have done a nice job of responding to many of the reviewers' comments. The paper is well written and the overall findings are impactful. One downside is the lack of physiological in vivo data with shCAD.

In several cases, the authors state that ammonia is 'potentially toxic to cancer cells'. However, there is growing evidence that ammonia is not toxic to cancers, but is efficiently assimilated into amino acids or nucleotides in some tumors, depending on tumor type and oncogenic drivers (papers by Gottlieb, Haigis, Kaelin, and Kalaany). Thus, this should be toned down in the text.

Reviewer #2:

Remarks to the Author:

The authors have done a reasonably good job of responding to the reviewers' comments. I am still somewhat puzzled by the accumulation of OA and I think the explanation offered could do with some clarification. It's not clear why the authors don't look for excreted OA.

I think there is a typo in the legend to Figure 5k, which says the cells are under normoxia – I think this is hypoxia. It would be useful to show the key for the color coding again here.

Does the aKG rescue affect glutamine uptake and metabolism?

Reviewer #1 (Remarks to the Author):

The authors have done a nice job of responding to many of the reviewers' comments.

The paper is well written and the overall findings are impactful. One downside is the lack of physiological *in vivo* data with shCAD.

Response: Thanks for your positive evaluations.

We add some discussions about the lack of physiological *in vivo* data with shCAD in the last paragraph of the Discussion section: “Our *in vitro* data showed that knockdown of CAD increased ammonia secretion from cancer cells under hypoxia. Although the *in vivo* tumor cells often grow in a hypoxic microenvironment, whether their ammonia production will be affected by CAD knockdown remains unclear due to the inefficient tumor formation of cancer cells with shCAD. However, we indeed measure the increased level of blood dihydroorotate/urotate in cancer patients or tumor-bearing mice.”

In several cases, the authors state that ammonia is 'potentially toxic to cancer cells'. However, there is growing evidence that ammonia is not toxic to cancers, but is efficiently assimilated into amino acids or nucleotides in some tumors, depending on tumor type and oncogenic drivers (papers by Gottlieb, Haigis, Kaelin, and Kalaany). Thus, this should be toned down in the text.

Response: Thanks for your suggestion. I agree with you. We already toned down the description about the toxicity of ammonia. There are six descriptions about ammonia toxicity throughout the paper.

1. In the abstract, “However, under such a condition how glutamine nitrogen is disposed to avoid releasing potentially toxic ammonia remains to be determined.” We changed this sentence to “However, under such a condition how glutamine nitrogen is disposed to avoid over-accumulating ammonia remains to be determined.”

2. In the third paragraph of the introduction section, we kept the sentence of “Concomitantly with these processes, the increasing amount of ammonia is produced and could be toxic to cells^{12,13}”, because it is a summary of the related references.

3. In the third paragraph of the introduction section, we changed the sentence of “The best way for proliferating cancer cells is to reduce the generation of potentially toxic ammonia” to “To avoid over-accumulating ammonia, the best way for proliferating cancer cells is to reduce its generation .”

4. In the first paragraph of the Discussion section, we keep the sentence of “The accumulation of these “by-products”, particularly ammonia, could be toxic to cells^{12,13,21}”, because it is also a summary of the related references.

5. In the second paragraph of the Discussion section, we change “The accumulated ammonia is potentially toxic to cells and should be safely removed” to “The accumulating ammonia should be safely removed”.

6. In the last paragraph of the Discussion section, we change “supporting that the excretion of dihydroorotate/urotate is an alternative pathway to dispose of the toxic ammonia” to “supporting that the excretion of dihydroorotate/urotate is an alternative pathway to dispose of ammonia”.

Thank you again for the constructive suggestions.

Reviewer #2 (Remarks to the Author):

The authors have done a reasonably good job of responding to the reviewers' comments. I am still somewhat puzzled by the accumulation of OA and I think the explanation offered could do with some clarification. It's not clear why the authors don't look for excreted OA.

Response: Thanks for your positive evaluations.

I realize that we made a mistake in responding to Question#4 about OA in the last response letter by “we did not measure the excretion of OA”. This explanation possibly puzzles this reviewer and makes this reviewer think we do not look for excreted OA. In fact, we want to convey the meaning of “we measured both OA and DHOA in the medium under hypoxia but no excreted OA was detected”. Hypoxia induces the accumulation of DHOA and OA in cells, but only DHOA, not OA, is detected in the medium (Figure 5c). Now we highlight the related description in the context: “However, we observed a significantly increased amount of dihydroorotate, but not orotate, in hypoxic medium of various cell lines, such as MCF-7, HeLa, A549, HCC-LM3, SGC-7901 and 4T1 (Fig. 5c), suggesting it could be an universal phenomenon responding to hypoxia” in the second paragraph of “Association of glutamine-carbon metabolism with its nitrogen assimilation under hypoxia” in Results section. Sorry for the last misguiding response.

I think there is a typo in the legend to Figure 5k, which says the cells are under normoxia – I think this is hypoxia. It would be useful to show the key for the color coding again here.

Response: Thanks for your carefully reading our manuscript. Yes, it is a typo in Figure 5k legend, and we already correct normoxia to hypoxia. We also add the color coding to Figure 5k.

Does the aKG rescue affect glutamine uptake and metabolism?

Response: Thanks for your suggestions.

We add new data to show that α -ketoglutarate supplementation suppressed hypoxia-promoted glutamine uptake (Extended Data Figure 12d). Correspondingly, we add the description in the related content in the last paragraph of “Association of glutamine-carbon metabolism with its nitrogen assimilation under hypoxia” in Results section. However, due to the interchanging between glutamine-derived glutamate/ α -ketoglutarate and exogenous α -ketoglutarate, it is very difficult (or impossible) to determine the effects of α -ketoglutarate supplementation on glutamine metabolism by tracing ^{13}C -labeled glutamine. Hopefully, you can understand this. Thank you so much again.